# Effects of a combined energy restriction and vigorous-intensity exercise intervention on the human gut microbiome: A randomised controlled trial

Russell G. Davies[1]  , Laura A. Wood[1], Aaron Hengist[1] , Ciara O'Donovan[2,3], Wiley Barton[2,3] ,
Fiona Crispie[3], Jean-Philippe Walhin[1] , Maria A. Valdivia-Garcia[4], Isabel Garcia-Perez[4], Gary Frost[4],
Orla O'Sullivan[2,3], Paul D. Cotter[2,3], Javier T. Gonzalez[1], James A. Betts[1], Françoise Koumanov[1]
and Dylan Thompson[1]

[1]*Department for Health, Centre for Nutrition Exercise and Metabolism, University of Bath, Bath, UK*
[2]*APC Microbiome Ireland, University College Cork, Cork, Ireland*
[3]*Teagasc Food Research Centre, Moorepark, Cork, Ireland*
[4]*Section of Nutrition Research, Division of Digestive Diseases, Department of Metabolism, Digestion and Reproduction, Imperial College London, South Kensington Campus, London, UK*

Handling Editors: Kim Barrett & Stephen Keely

The peer review history is available in the Supporting Information section of this article (https://doi.org/10.1113/JP287424#support-information-section).

*The Journal of Physiology*

**Abstract figure legend** The human gut microbiome remains stable in the face of an intensive energy restriction and vigorous-intensity exercise intervention that improved body composition and metabolic health in people with overweight/obesity. Thus, early metabolic changes with weight loss in humans are unlikely to be mediated by changes to the gut microbiome.

**Abstract** Metabolic health improvements in response to exercise and energy restriction may be mediated by the gut microbiome, yet causal evidence in humans remains limited. We used a 3-week exercise and energy restriction intervention to examine changes to the gut microbiome in otherwise healthy sedentary men and postmenopausal women with overweight/obesity. Inter-

vention participants ($n = 18$) reduced habitual energy intake by 5000 kcal/week and expended 2000 kcal/week in addition to habitual physical activity through treadmill walking at 70% V̇$O_{2Peak}$. Control participants ($n = 12$) maintained their usual lifestyle. Participants underwent dual-energy X-ray absorptiometry (DEXA), and samples of faeces, fasted venous blood, subcutaneous adipose tissue and skeletal muscle were collected. Faecal DNA was sequenced and profiled using shotgun metagenomics, Kraken2/Bracken and Human Microbiome Project Unified Metabolic Analysis Network 2 (HUMAnN2). The intervention significantly reduced body mass (mean $\Delta \pm$ SD: –2.6 $\pm$ 1.5 kg), fat mass (–1.5 $\pm$ 1.3 kg), fasted insulin (–23.5 $\pm$ 38.1 pmol/l), leptin (–10.6 $\pm$ 7.3 ng/ml) and total cholesterol (–0.70 $\pm$ 0.42 mmol/l) concentrations, and also improved insulin sensitivity (HOMA2%S (homeostatic model of assessment)). Despite these significant metabolic changes the gut microbiome was unchanged in terms of $\alpha$ and $\beta$ diversity and relative abundance. Thus, despite clinically meaningful improvements in body composition and metabolic health, we found no evidence for changes to the gut microbiome. In conclusion early metabolic changes with weight loss in humans are unlikely to be mediated by changes to the gut microbiome.

(Received 29 September 2024; accepted after revision 9 July 2025; first published online 14 August 2025)

**Corresponding author** Dylan Thompson: Department for Health, Centre for Nutrition Exercise and Metabolism, University of Bath, Bath, Somerset, UK.    Email: d.thompson@bath.ac.uk

**Key points**

- Changes to the gut microbiome could contribute to metabolic improvements associated with weight loss in humans, but there have been limited attempts to address this question using robust randomised controlled trials (RCTs).
- We used a parallel-group RCT to examine whether a 3-week combined energy intake restriction and vigorous-intensity exercise intervention in people with overweight and obesity was temporally associated with changes to gut microbiome taxonomic composition and functional potential, short-chain fatty acid concentrations and expression of genes related to host–microbiome interactions in skeletal muscle and subcutaneous adipose tissue.
- We found that the human gut microbiome remains unchanged in the face of an intensive energy intake restriction and vigorous exercise intervention that significantly improved body composition and metabolic health in people with overweight/obesity.
- These findings indicate that early metabolic changes with weight loss in humans are unlikely to be mediated by changes to the gut microbiome.

## Introduction

Obesity and overweight are major global public health issues, with a causal role in the aetiology of serious diseases such as type 2 diabetes and associated premature mortality (Tancredi et al., 2015). Positive energy balance associated with high dietary energy intake and/or low physical activity is a primary factor contributing to excess adiposity, so addressing these lifestyle factors is the most common form of treatment (Hruby et al., 2016). It is widely recognised that the gut microbiome is integral to human health (Lynch & Pedersen, 2016), and it has been proposed that the composition of the gut microbiome plays a role in the aetiology of obesity and metabolic

**Russell G. Davies** studied sport and exercise science at the University of Bath between 2013 and 2017, and subsequently received his PhD from the University of Bath in 2023. His doctoral research investigated the interactions between physical activity, nutrition, metabolism and the human gut microbiome, and involved establishing collaborations with APC Microbiome Ireland and Imperial College London. Russell is currently working as a postgraduate lecturer at the Eastern Institute of Technology (EIT) in Napier, New Zealand. He continues to collaborate with colleagues at the University of Bath to publish the findings from his PhD research.

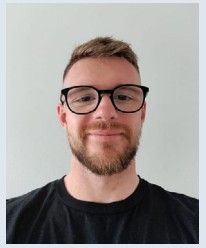

syndrome through the regulation of insulin sensitivity and adipose tissue accumulation (Canfora et al., 2015; Canfora et al., 2019). The gut microbiome is a complex ecosystem, the composition and function of which are affected by a range of factors, including host genetics (Bonder et al., 2016; Kurilshikov et al., 2021) and environmental exposures, such as diet (David et al., 2014) and medication (Maier et al., 2018). Physical activity (Lin et al., 2015) and weight loss (Mora-Rodriguez et al., 2018) are both potent stimuli for improvements in insulin sensitivity and cardio-metabolic health, and it has been suggested that these effects may be partially mediated by changes in the gut microbiome (Aragón-Vela et al., 2021; Sohail et al., 2019).

Previous evidence from rodent studies (Allen et al., 2015; Fernández et al., 2021), observational studies (Barton et al., 2018; Clarke et al., 2014) and human interventions (Allen et al., 2018; Kern et al., 2020) suggests that regular exercise may modulate the composition of the gut microbiome. In addition, weight loss has been associated with increased gut microbiome $\alpha$ diversity and reduced markers of intestinal permeability in humans; however, much of this evidence is from observational studies or within group analyses, which limits causal inference (Koutoukidis et al., 2022). Thus the possibility that exercise and weight loss may affect metabolic health via the gut microbiome is largely unexplored in the studies published to date.

The potential mechanisms whereby exercise and weight loss may exert positive effects on insulin sensitivity via the gut microbiome include the increased production of short-chain fatty acids (SCFAs) and activation of G-protein coupled receptors (GPR) 41 and GPR43 (also known as free fatty-acid receptors (FFARs) FFAR3 and FFAR2) in the peripheral tissues responsible for the disposal of ingested nutrients, such as skeletal muscle and subcutaneous adipose tissue (Canfora et al., 2015). Little is known about the effects of exercise and weight loss on these pathways. Two previous exercise intervention studies measured faecal SCFA (Allen et al., 2018; Liu et al., 2020), and one study measured SCFAs in peripheral circulation (Liu et al., 2020). To the best of our knowledge, no exercise or weight loss interventions have measured the expression of their receptors in metabolically important tissues. Another mechanism could involve changes to the translocation of lipopolysaccharide (LPS) from the gut lumen. Increased circulating LPS and associated inflammation and metabolic dysfunction have been termed 'metabolic endotoxaemia' (Cani et al., 2007). Exercise training reduces systemic inflammation (Thompson et al., 2010) (Gonzalo-Encabo et al., 2021), but there is limited evidence from human trials investigating whether this is linked to changes in the gut microbiome and metabolic endotoxaemia. Thus, little is known about whether improvements in host metabolism with energy restriction and/or exercise are linked to changes in key pathways such as SCFA-mediated activation of FFAR2/FFAR3 and/or a reduction in metabolic endotoxaemia.

To determine whether metabolic improvements associated with exercise and weight loss are associated with changes in the gut microbiome in the short term, we conducted a randomised controlled trial (RCT) using an interventional protocol known to improve metabolic health (Walhin et al., 2016). We investigated the effects of combined vigorous-intensity exercise and dietary energy restriction (without modification to diet composition) in a group of otherwise healthy sedentary adults with overweight or obesity who were at risk of developing insulin resistance. We characterised the effects of the intervention on body composition, insulin sensitivity, systemic inflammation, gut microbiome taxonomic composition and functional potential, levels of SCFA in faeces and serum and expression of key genes related to host–microbiome interactions in both skeletal muscle and subcutaneous adipose tissue. We hypothesised that the expected significant improvements in metabolic health after the intervention would be accompanied by significant changes in gut microbiome indices compared to the control group.

## Methods

### Participants

Participants were recruited from the community through local advertisement and the Primary Care Research Network. Participants were required to be between the ages of 40 and 65 years at the point of study entry; have a body mass index (BMI) $\geq$25 and $\leq$40 kg/m$^{-2}$, a waist circumference >80 cm (females) and >94 cm (males) and blood pressure $\leq$160/100 mmHg (systolic/diastolic); and have been weight stable ($\pm$2.5%) for >3 months. Female participants were required to be postmenopausal (no menstruation for at least 1 year) (Witteman et al., 1989). Individuals who smoked within the past 12 months, took antibiotics within the past 3 months, suffered from a diagnosed medical condition with contraindications for exercise or took medication likely to affect study measures or pose undue personal risk were excluded from participating. Antibiotic use is common, and therefore 3 months was selected as a reasonable antibiotic-free period to allow time for the microbiome to normalise without excluding otherwise eligible participants from a study with a large number of inclusion/exclusion criteria. No participants started a course of antibiotics during their involvement in the study. Participants completed a health questionnaire and Physical Activity Readiness Questionnaire (PAR-Q) to verify their eligibility and provided written informed consent prior to participating.

**Table 1. Characteristics and typical dietary intake of control and intervention participants at baseline.**

|  | Control (*n* = 12) | Intervention (*n* = 18) |
|---|---|---|
| **Participant characteristics** | | |
| Female | 9 (75%) | 11 (61%) |
| Male | 3 (25%) | 7 (39%) |
| Age (year) | 59 (5) | 57 (6) |
| Height (m) | 1.66 (0.07) | 1.69 (0.10) |
| Body mass (kg) | 86.7 (14.8) | 91.0 (14.0) |
| Body mass index (kg/m$^2$) | 31.5 (5.4) | 31.7 (2.8) |
| Waist circumference (cm) | 99 (13) | 105 (9) |
| Systolic blood pressure (mmHg) | 128 (9) | 134 (15) |
| Diastolic blood pressure (mmHg) | 79 (8) | 86 (7) |
| Peak oxygen uptake (ml/kg/min) | 24.0 (4.7) | 24.0 (4.5) |
| Total Energy expenditure (kcal/day) | 2537 (512) | 2702 (545) |
| Physical activity level | 1.61 (0.09) | 1.61 (0.11) |
| **Dietary intake** | | |
| Total energy (kcal/day) | 2111 (544) | 2221 (635) |
| Carbohydrate (g/day) | 225 (62) | 225 (70) |
| Protein (g/day) | 85 (18) | 86 (20) |
| Fat (g/day) | 91 (34) | 90 (32) |
| Alcohol (g/day) | 8 (10) | 17 (18) |
| Fibre (g/day) | 24 (7) | 24 (9) |

*Note*: Data are count (%) and mean (SD). Dietary intake data are from seven continuous days of weighed food and fluid record.

To focus on individuals typical of the sedentary population for whom a physical activity intervention would be recommended, participants were included only after confirming they had a daily average physical activity level (PAL) ≤1.75 using individually calibrated combined heart rate and accelerometery (Thompson et al., 2006). Participant characteristics at baseline are presented in Table 1.

### Experimental design and protocol

A randomised parallel-group design was used in the current study. We used a combined exercise and caloric restriction intervention that we previously employed, and data from this previous study were used to determine the necessary sample size (Walhin et al., 2016). It was estimated that a sample size of 24 (16 in intervention and 8 in control) would provide approximately 90% power to detect a reduction in fasting plasma insulin of −19.9 ± 18.6 pmol/l in in the intervention group *versus* 0 ± 4.3 pmol/l in control using an independent *t* test at an *α* level of 0.05. This sample size was rounded up to 20 in the intervention group and 10 in control, giving a total requirement of 30 participants to be recruited on a rolling basis. Participants were randomised by a non-participant facing member of the research team (1:2 allocation ratio, control to intervention), and randomisation was stratified by sex and waist circumference as predictors of insulin resistance with

cut-offs of 102 and 88 cm for men and women, respectively (Wahrenberg et al., 2005). A group allocation ratio of 1:2 was chosen to accommodate the anticipated greater variability and potentially greater dropout in the intervention group. A CONSORT diagram showing participant flow through the trial is shown in Fig. 1. Participants were instructed to either maintain their current lifestyle (control) or induce a fixed energy deficit of 29,302 kJ/week (7000 kcal/week) through dietary energy restriction and increased vigorous-intensity physical activity (intervention) for 3 weeks (Walhin et al., 2016). Participants in the intervention group were required to under-consume their habitual diet by 20,930 kJ/week (5000 kcal/week) and to complete five treadmill walking sessions per week at 70% of their peak oxygen uptake (%$\dot{V}O_{2Peak}$) to expend 1674 kJ (400 kcal) per session above rest (i.e. 2000 kcal/week).

All participants were asked to maintain their habitual lifestyle and dietary habits prior to the baseline laboratory visit, with the exception of avoiding caffeine and alcohol 24 h before testing. For baseline testing participants arrived at the laboratory at 8:00 AM ± 1 h after an overnight fast (≥10 h). Body mass was measured using electronic scales (Tanita Co., Tokyo, Japan), and waist circumference was measured to the nearest 0.5 cm using a tape measure (SECA, Hamburg, Germany). Body composition was determined using dual-energy X-ray absorptiometry (DEXA, Discovery, Hologic, Marlborough, Massachusetts, USA). Resting metabolic

rate (RMR) was assessed using the Douglas bag technique in a semi-supine position through repeated 5-min expired air samples over ~30 min in accordance with best practice (Compher et al., 2006). A fasted baseline blood sample, a subcutaneous adipose tissue biopsy and a percutaneous muscle biopsy were then obtained. Blood pressure was measured in triplicate before participants left the laboratory. All procedures were repeated at follow-up 22–26 days after baseline testing.

### Preliminary measurements

After screening, participants completed a weighed food record for a representative 7-day period using a digital scale (Smart Weigh Zhongshan, China). Diet records were analysed using Nutritics online software (Nutritics Research Edition, version 5.031, Swords, Ireland) to estimate energy intake, and pre-intervention diet-induced thermogenesis (DIT) was estimated as 10% of energy intake (Westerterp, 2004). Combined heart rate/accelerometery (Actiheart 4, CamNtech, Fenstanton, UK) with an individual calibration was used to determine habitual physical activity energy expenditure over the same period as diet monitoring (Thompson et al., 2006). After the diet and physical activity monitoring period,

participants reported to the laboratory at 9:00 AM $\pm$ 1 h after an overnight fast ($\geq$10 h). RMR was measured before participants completed a peak oxygen uptake ($\dot{V}O_{2Peak}$) test on a treadmill (Woodway, ELG 70, Waukesha, WI, USA) using a protocol adapted from Taylor et al. (1955). Expired air was collected and analysed using the Douglas bag method after fluctuations in inspired air composition was accounted for (Betts & Thompson, 2012). Energy expenditure was calculated based on standard stoichiometric equations assuming negligible contribution of protein (Frayn, 1983; Jeukendrup & Wallis, 2005).

### Energy restriction and exercise intervention

For each intervention participant a weekly target energy intake was calculated by subtracting 5000 kcal from their estimated total energy expenditure over 7 days (estimated via Actiheart). Target energy intake estimates were based on estimated energy expenditure rather than reported energy intake as under-reporting of energy intake is common (Macdiarmid & Blundell, 1998). Reported total energy intake over the 7 days of recorded diet was then used to calculate a reduction factor to scale down the amount of each diary entry to meet the target energy

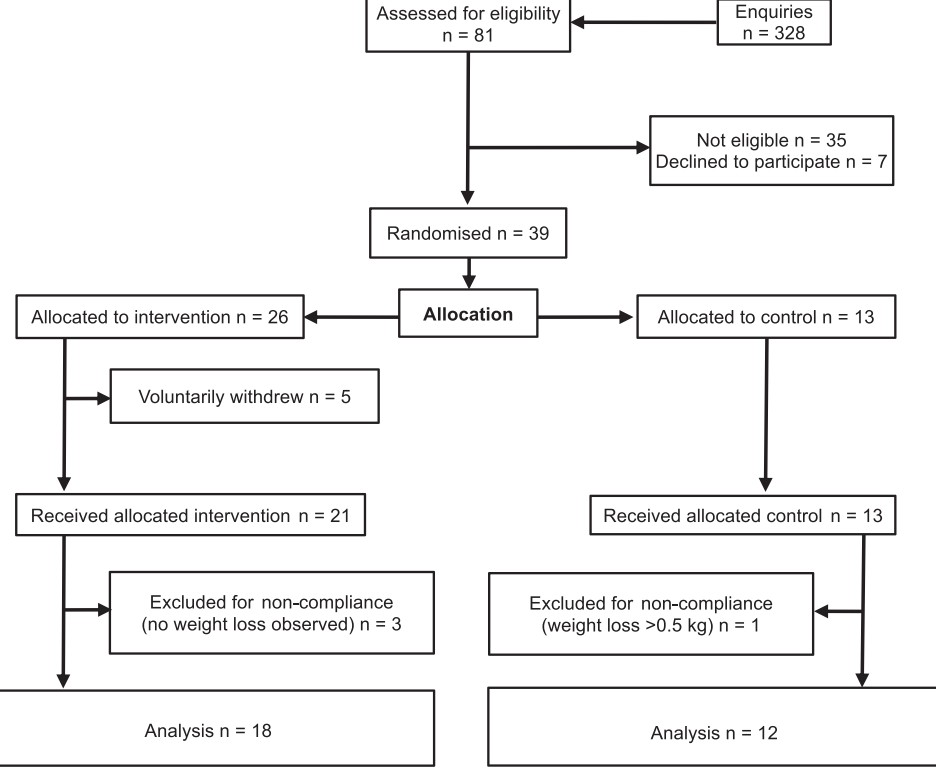

**Figure 1. CONSORT diagram showing participant flow through the trial**
Non-compliance defined as lack of weight loss in the intervention group and weight loss >0.5 kg in the control group. The one participant excluded from the control group lost >2 kg (they self-reported an illness during the 3-week period).

intake (Walhin et al., 2016). Participants had their adjusted diet diary returned to them before beginning the intervention and were instructed to replicate their 7 days of recorded intake for three consecutive weeks, consuming the specified reduced amount of each item. In this way habitual diet composition was not changed, whereas the total amount of food and therefore energy consumed were reduced.

As a hypothetical worked example of the energy restriction calculation, if a participant recorded a total weekly energy intake of 20,000 kcal and a total weekly energy expenditure of 21,000 kcal, his or her target weekly energy intake during the intervention period would be 16,000 kcal (21,000 – 5000 kcal) (Walhin et al., 2016). Dividing their target weekly energy intake during the intervention (16,000 kcal) by their recorded energy intake (20,000 kcal) gives a value of 0.8, indicating that the weight of each food item recorded in their food and fluid record should be reduced by 20% to elicit the intended energy deficit during the intervention period.

Participants in the intervention group were also required to complete five sessions of treadmill walking per week at the speed and gradient estimated to require at least 70% $\dot{V}O_{2Peak}$ based on the results from the incremental exercise test completed during the preliminary measures. The method of prescribing exercise intensity and duration has been described previously (Walhin et al., 2016), whereby a set total distance per session was calculated for each participant to expend 1674 kJ (400 kcal) above resting energy expenditure. Exercise training was monitored through a weekly supervised session, and participants were instructed to record their heart rate during every session (supervised and unsupervised) using chest-worn monitors with inbuilt memory (TickrX, Wahoo, Atlanta, GA, USA). Intervention participants verbally confirmed that they had consumed only the diet prescribed during the intervention period and had completed all exercise sessions. The final exercise bout was performed at least 36 h before follow-up testing.

## Exercise adherence

All participants reported completing all exercise sessions as instructed. Due to technical issues, continuous heart rate data were recorded for 53% of sessions and, of those sessions recorded, heart rate was 107 ± 8% (mean ± SD) of the target, indicating that the majority of sessions were completed at or above the required intensity of 70% $\dot{V}O_{2Peak}$.

## Blood sampling and analysis

During laboratory visits at baseline and follow-up, an 18-gauge cannula (Venflon Pro, BD, Franklin Lakes, NJ, USA) was inserted into an antecubital vein, and a fasted blood sample was collected. Five millilitres of whole blood was dispensed into an EDTA acid-coated tube (Sarstedt AG & Co. KG, Nümbrecht, Germany), and 5 ml was dispensed into a serum-separator tube (Sarstedt AG & Co. KG, Nümbrecht, Germany). Plasma tubes were immediately centrifuged, whereas serum tubes were left to clot at room temperature for 15 min before centrifugation. Plasma and serum tubes were centrifuged at 2500 *g* for 10 min at 4°C. After centrifugation, plasma and serum were immediately aliquoted and frozen on dry ice, and then stored at –80°C until subsequent analysis.

Plasma glucose, total cholesterol and high-density lipoprotein (HDL) cholesterol concentrations were analysed using commercially available kits on a Daytona-automated clinical chemistry analyser (Randox, Crumlin, UK). Serum insulin (Mercodia, Uppsala, Sweden) and leptin and lipopolysaccharide binding protein (LBP) (R&D Systems, Minneapolis, MN, USA) concentrations were determined using enzyme-linked immunosorbent assays (ELISA). C-reactive protein (CRP) was analysed using electrochemiluminescence assay (Meso Scale Diagnostics LLC, Rockville, MD, USA). All assays were conducted according to the manufacturer's instructions.

## Adipose tissue biopsy and processing

An adipose tissue biopsy was obtained from the abdomen 4–7 cm lateral of the umbilicus under local anaesthesia (1% lidocaine) with a 14-gauge needle using the needle aspiration technique (Walhin et al., 2013). Follow-up samples were taken from the opposite side of the abdomen from baseline. The sample was cleaned with saline, and blood clots were manually removed before aliquoting (Trim et al., 2022). Portions of samples were immediately snap frozen in liquid nitrogen and stored at –80°C before subsequent analysis.

## Muscle tissue biopsy and processing

Under local anaesthetic (1% lidocaine) a 3–5 mm skin incision was made to the anterior aspect of the thigh using a surgical blade. A percutaneous muscle sample was then obtained from the vastus lateralis using the suction-modified needle biopsy technique (Tarnopolsky et al., 2011). Follow-up samples were obtained from the same leg >3 cm away from the first sample site. Muscle samples were collected from a subset of participants ($n = 9$ from the control group and $n = 13$ from the intervention group).

## Quantitative real-time PCR

Adipose and muscle samples were prepared for quantitative real-time PCR as described (Walhin et al., 2013), and TaqMan qPCR assays from Applied Biosystems (Warrington, UK) were used to assess the expression of FFAR2/GPR43 (Hs00271142_s1), FFAR3/GPR41 (Hs04937136_g1), LBP (Hs01084628_m1) and CD14 (Hs00169122_g1). Housekeepers were peptidylprolyl isomerase A (PPIA, Hs99999904_m1) for adipose and PPIA (Hs99999904_m1), GAPDH (Hs99999905_m1) and $\beta$-actin (ACTB, Hs99999903_m1) for muscle.

## Stool and urine sample collection and processing

Participants provided a urine sample on each trial day, $\sim$4 ml of which was aliquoted and immediately frozen on dry ice and stored at –80°C for subsequent analysis. Participants were provided with a collection kit and instructed to collect a stool sample within the 24 h prior to visiting the laboratory for each trial day. Some participants were unable or unwilling to provide faecal samples, and therefore microbiome data are presented for $n = 23$ of 30 participants (9 control and 14 intervention). Faecal collection kits included faeces catcher paper (Abbexa, Cambridge, UK) and a sterile collection container (Sarstedt AG & Co. KG, Nümbrecht, Germany). In accordance with published recommendations, samples were stored at $\sim$4°C in a domestic refrigerator and transported to the laboratory in thermally insulated bags with frozen ice packs (Gratton et al., 2016) before being processed and stored at –80°C immediately after collection and within 24 h. Stool samples were manually homogenised using a sterile disposable spatula (Corning Inc., Corning, NY, USA). To minimise thawing of stool samples prior to the addition of lysis buffer during DNA extraction, $\sim$200 mg of homogenised stool was aliquoted directly into autoclaved bead-beating tubes (Sarstedt AG & Co. KG, Nümbrecht, Germany) and stored at –80°C.

## SCFA measurement in serum and faeces

SCFA levels in serum and faeces were determined at Imperial College London as described previously (Valdivia-Garcia et al., 2022). Briefly $\sim$50 mg of stool was weighed into a bead-beating tube with 100 µl of ice-cold isopropanol and 100 mg of 1 mm glass beads. Tubes were beaten for 90 s at 6000 rpm in a Precellys Evolution homogeniser (Bertin Technologies, Montigny-le-Bretonneux, France) and then centrifuged at 16,000 *g* for 15 min at 4°C, and 5 µl of the supernatant was transferred to a new tube containing 45 µl of HLPC-grade water. For serum, 50 µl was added to 100 µl of ice-cold isopropanol and vortexed, and then centrifuged at 16,000 *g* for 15 min at 4°C; then 50 µl was added to a new

tube. Internal standards (acetic acid-$^{13}C_2$, propionic acid-1-$^{13}C$ and butyric acid-1,2-$^{13}C_2$), 20 µl of 200 mM 3-nitrophenylhydrazine hydrochloride and 20 µl of 120 mM (*N*-(3-dimethylaminopropyl)-ethylcarboiimide) hydrochloride were added to the tubes containing supernatant, and the tubes were vortexed and incubated for 30 min at 40°C. Samples were removed from the heat block, and 200 µl of 0.1% formic acid was added; then the samples were vortexed and transferred to vials for LC injection. Ultra-high performance liquid chromatography with tandem mass spectrometry (UPLC-MS/MS) was performed as described previously (Valdivia-Garcia et al., 2022). Calibration curve $r^2$ values were 0.999 for all SCFAs. Limits of detection were 0.16, 0.04 and 0.26 µM for acetate, propionate and butyrate, respectively. Limits of quantitation were 0.48, 0.13 and 0.78 µM for acetate, propionate and butyrate, respectively. Samples with concentrations lower than the limit of quantitation were excluded from statistical analysis.

## Shotgun metagenomic sequencing and data processing

Total DNA was extracted from frozen human stool samples using the QIAamp Fast DNA Stool Mini Kit (category number 51,504, Qiagen, Hilden, Germany) according to the manufacturer's instructions with the addition of a bead-beating step (Lim et al., 2018). For bead beating $\sim$200 mg of stool was transferred to a sterile 2 ml screw-cap tube containing 0.25 g of zirconia beads (0.125 g of 0.1 mm, 0.125 g of 1.0 mm and a single 2.5 mm bead, category numbers 11079101z-BSP, 11079110z-BSP and 11079125z-BSP, respectively). Tubes were placed in a Precellys Evolution homogeniser (Bertin Technologies, Montigny-le-Bretonneux, France) and underwent bead beading for 3 min at 6000 rpm.

Extracted faecal DNA was prepared for sequencing using the Nextera XT DNA Library Preparation Kit (category FC-131-1096, Illumina Inc., San Diego, CA, USA) according to the manufacturer's instructions with the exception that the tagmentation time was increased to 7 min. Tagmentation and amplification were completed on an Applied Biosystems 2720 thermal cycler (Thermo Fisher Scientific Inc., Waltham, MA, USA) using XT combinatorial indices as described by the manufacturer. After amplification and Ampure clean-up of the libraries as per the Illumina guidelines, fragment size ranges of individual samples were checked using the Agilent High Sensitivity DNA Kit (category 5067-4626, Agilent Technologies, Santa Clara, CA, USA) on the Agilent 2100 Bioanalyzer (category G2939BA, Agilent Technologies, Santa Clara, CA, USA). The DNA concentration of each sample was also determined using the Qubit dsDNA high-sensitivity assay (category Q32851, Thermo Fisher Scientific Inc., Waltham, MA, USA) on the

Qubit 4 Fluorometer (category Q33238, Thermo Fisher Scientific Inc., Waltham, MA, USA). Based on these metrics the libraries were pooled at equal molarity, and the concentration of the final pooled libraries was determined by qPCR using the KAPA Library Quantification Kit (category KK4824, Kapa Biosystems, Wilmington, MA, USA) on the LightCycler 480 (category 05015278001, Roche, Basel, Switzerland). The pooled libraries were sequenced using the $2 \times 150$ bp High Output Kit (category 20024908, Illumina Inc., San Diego, CA, USA) on the Illumina NextSeq 500 sequencing system according to the manufacturer's instructions.

Contaminating host (human) reads were removed using a reference library created in Bowtie2 (version 2.3.4) (Langmead & Salzberg, 2012). Resulting FastQ files were trimmed and quality filtered using Trim Galore (phred $= 33$, minimum length $= 20$). Taxonomic classifications of trimmed reads were determined using Kraken2 and Bracken using the genome taxonomy database (GTDB) (database downloaded: 30 June 2020) (Lu et al., 2017; Wood et al., 2019). Functional profiling was completed using the Human Microbiome Project Unified Metabolic Analysis Network 2 (HUMAnN2) (Franzosa et al., 2018). A total of $3.64 \times 10^8$ trimmed reads were used for classification after contaminant removal, averaging $7.92 \times 10^6$ reads per sample. Data for this study are available at the European Nucleotide Archive, https://www.ebi.ac.uk/ena/browser/ (data accession ID: PRJEB80390).

### Statistical analysis

Descriptive statistics were calculated, statistical tests were conducted, and figures were drawn using R (version 4.1.0) in RStudio (version 2022.07.1+554). In accordance with current recommendations for pre-post randomised study designs, analysis of covariance (ANCOVA) was used to determine the effects of the intervention on continuous outcome variables, with follow-up values as the dependent variable, group allocation as the independent variable and baseline values included as a covariate (Wan, 2021). ANCOVA provides the most unbiased, precise and efficient causal estimates for continuous outcomes from parallel-group RCTs (Bland & Altman, 2015). Insulin sensitivity was assessed through the updated homeostatic assessment model (HOMA2) (Levy et al., 1998) using the HOMA2 tool (Hill et al., 2013). Where deviations from normality were detected using the Shapiro–Wilk test, continuous variables were log transformed prior to analysis. Gene expression data were analysed using independent $t$ tests on the log-transformed delta–delta cycle threshold (Ct) data after normalisation to housekeeper genes (cyclophilin for adipose, geometric mean of PPIA, GAPDH and $\beta$-actin for muscle). Statistical significance was accepted at $P \leq$ 0.05. Data for the study are available at the University of Bath Research Data Archive (Koumanov et al., 2025).

Microbiome taxonomic composition data were analysed at the genus and species levels, and functional potential was analysed using normalised pathway abundances not stratified by species. Output tables of raw reads were filtered to remove taxa that were not present in at least 5% of samples, and the *PERFect* permutational filtering R tool was then used to remove low-signal taxa (Smirnova et al., 2018). $\alpha$ diversity was calculated at the species level using the Shannon index to account for richness and evenness (Park & Allaby, 2017). $\beta$ diversity analysis was conducted at the species level based on robust Aitchison distances to account for the sparse and compositional nature of microbiome data (Martino et al., 2019). Non-metric multidimensional scaling (NMDS) plots were used to visualise $\beta$ diversity. Permutational multivariate analysis of variance (PERMANOVA) was used to test for differences in the $\beta$ diversity of species and functional pathways between groups at each time point and between time points within each group, with permutations constrained to the level of participant where required to account for repeated measures sampling. $\alpha$ and $\beta$ diversity analyses were conducted using the *vegan* R package, version 2.6-2 (Oksanen et al., 2022). Differential abundance of taxa and functional pathways within and between each group was assessed using ALDEx2 and expressed as estimated effect size of change per taxon and per pathway (Fernandes et al., 2014). Where multiple comparisons were made, *P*-values were adjusted using the Benjamini–Hochberg method, and statistical significance was accepted at $q \leq 0.1$.

## Results

### Anthropometric and physiological measures

Total body mass, fat mass, fat-free mass and waist circumference were significantly lower at follow-up in the intervention group compared to control (Table 2).

### Fasted blood measurements

Fasted serum insulin and plasma leptin, total cholesterol and low-density lipoprotein (LDL) cholesterol were all significantly lower in the intervention group at follow-up compared to control (Table 2). Fasting plasma glucose, HDL cholesterol, CRP and LBP were not significantly different at follow-up (Table 2).

### Insulin sensitivity

Insulin sensitivity as estimated by the interactive homeostatic model of assessment (HOMA2%S) was significantly higher, and $\beta$-cell activity (HOMA2%B) was significantly

**Table 2. Body composition and fasted blood measurements for each group at baseline and follow-up with change and *P*-values.**

| | Control (*n* = 12) | | | Intervention (*n* = 18) | | | ANCOVA *P*-value |
|---|---|---|---|---|---|---|---|
| | Pre | Post | Change | Pre | Post | Change | |
| Body mass (kg)[2] | 86.7 (14.8) | 87.2 (15.0) | 0.57 (0.84) | 91.0 (14.0) | 88.4 (14.3) | −2.6 (1.5) | <0.001 |
| Fat mass (kg)[2] | 34.0 (9.8) | 34.2 (9.9) | 0.2 (1.0) | 34.3 (5.7) | 33.0 (5.9) | −1.5 (1.3) | 0.00100 |
| Fat-free mass (kg)[2] | 49.1 (9.8) | 49.3 (10.2) | 0.2 (0.9) | 53.9 (11.6) | 52.0 (12.2) | −1.0 (1.9) | 0.0430 |
| Waist circumference (cm) | 99 (13) | 101 (10) | 2 (3) | 105 (9) | 101 (10) | −4 (3) | 0.00200 |
| Insulin (pmol/l) | 62.3 (54.0) | 66.5 (73.0) | 4.2 (24.1) | 69.5 (51.7) | 46.0 (25.0) | −23.5 (38.1) | 0.0340 |
| Glucose (mmol/l) | 5.79 (0.57) | 5.57 (0.42) | −0.22 (0.49) | 5.82 (0.56) | 5.74 (0.42) | −0.07 (0.40) | 0.195 |
| Leptin (ng/ml) | 39.0 (34.2) | 37.1 (31.7) | −1.9 (6.4) | 29.3 (17.8) | 18.7 (12.7) | −10.6 (7.3) | <0.001 |
| Total cholesterol (mmol/l) | 5.37 (1.00) | 5.28 (0.82) | −0.09 (0.37) | 5.85 (0.84) | 5.15 (0.80) | −0.70 (0.42) | <0.001 |
| HDL cholesterol (mmol/l) | 1.34 (0.27) | 1.32 (0.34) | −0.02 (0.13) | 1.39 (0.42) | 1.33 (0.42) | −0.06 (0.23) | 0.635 |
| LDL cholesterol (mmol/l)[1] | 3.43 (0.89) | 3.37 (0.74) | −0.06 (0.32) | 3.82 (0.65) | 3.29 (0.59) | −0.53 (0.37) | 0.00400 |
| Triglycerides (mmol/l) | 1.30 (0.46) | 1.29 (0.59) | −0.02 (0.31) | 1.40 (0.60) | 1.15 (0.30) | −0.26 (0.42) | 0.101 |
| HOMA2%S | 161 (180) | 172 (188) | 11.0 (41.3) | 114 (98.4) | 163 (146) | 48.7 (54.1) | 0.0160 |
| HOMA2%B | 75.8 (48.2) | 83.8 (59.6) | 8.01 (18.5) | 83.4 (44.0) | 63.1 (21.0) | −20.27 (30.7) | 0.00700 |
| LBP (ng/ml) | 7365 (3574) | 7108 (2985) | −257 (1512) | 7987 (3725) | 7008 (2759) | −979 (1709) | 0.257 |
| CRP (μg/ml) | 3.02 (3.28) | 2.61 (2.88) | −0.41 (0.73) | 3.93 (3.82) | 2.79 (2.29) | −1.14 (1.82) | 0.226 |

*Note*: Mean (SD). Significance values shown are ANCOVA effect of group.
[1] Estimated using the Friedewald equation.
[2] Due to technical issues DEXA data are available for *n* = 28 participants.
Abbreviations: ANCOVA, analysis of covariance; CRP, C-reactive protein; DEXA dual-energy X-ray absorptiometry; HDL, high-density lipoprotein; HOMA, homeostatic model of assessment; LBP, lipopolysaccharide binding protein; LDL, low-density lipoprotein.

lower at follow-up in the intervention group compared to control (Table 2).

### Gut microbiome α diversity

No significant differences in Shannon index were observed between the intervention and control groups at follow-up (Fig. 2; $P = 0.934$).

### Gut microbiome β diversity

Intra- and intergroup comparisons of species and functional metabolic pathway abundance β diversity are shown in Fig. 3. No significant effect of group was observed on species β diversity ($P = 0.101$), nor on metabolic pathway β diversity ($P = 0.639$). No significant differences in dispersion were observed between any group and time point combination.

### Relative taxonomic and functional pathway abundance

Inter- and intragroup comparisons of individual taxa at genus level are shown in Fig. 4. No genera were significantly changed in abundance between baseline and follow-up (all $q > 0.1$). Between groups, no genera were

significantly differentially abundant at either time point (all $q > 0.1$). The same analyses were also conducted at species level and at the functional metabolic level using normalised unstratified pathway abundance, and no species or functional pathways were significantly differentially abundant at these levels (all $q > 0.1$, Figs 5 and 6, respectively).

## Serum and faecal SCFA concentrations

Serum and faecal SCFA concentrations are shown in Figs 7 and 8, respectively. There was no significant effect on any of the serum and faecal SCFA concentrations measured (all $P > 0.05$). Urinary acetic, propionic and butyric acids and serum butyric acid concentrations were all found to be below the limit of quantitation using UPLC-MS/MS.

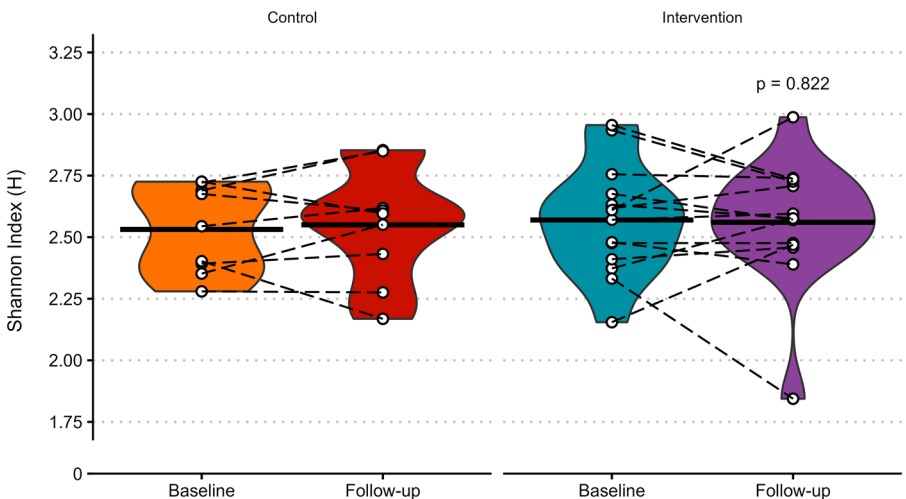

**Figure 2. α diversity of gut microbiome species assessed using the Shannon index**
Black crossbars represent the mean, white points show individual data with paired observations joined by dashed black lines and *P*-value shows ANCOVA (analysis of covariance) effect of group. *N* = 9 in control and 14 in intervention.

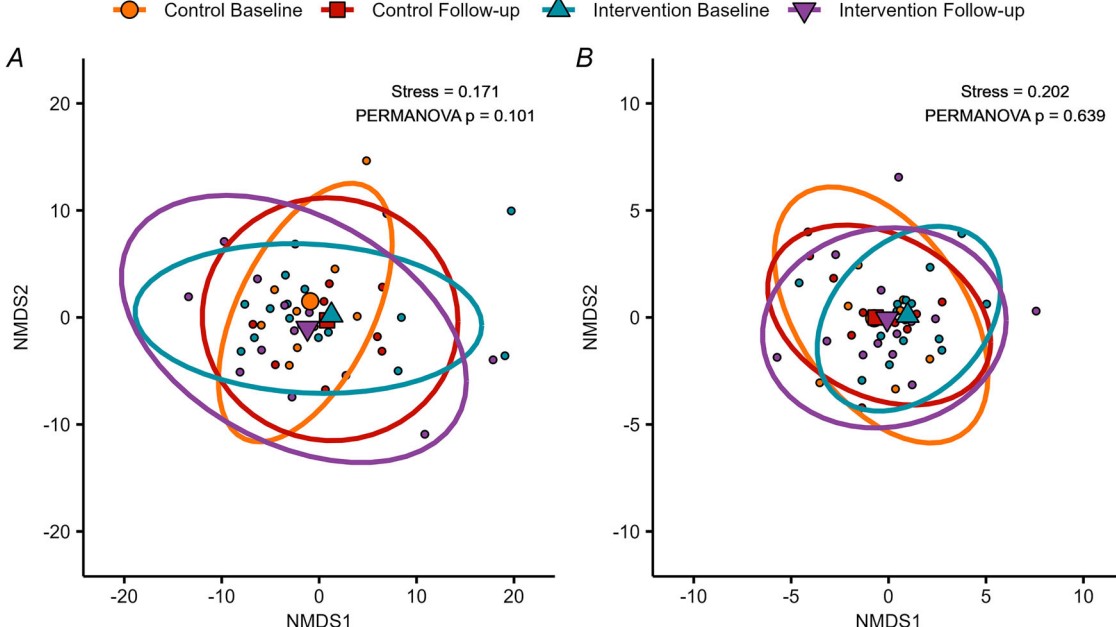

**Figure 3. β diversity plots of gut microbiome species (A) and normalised unstratified metabolic pathway (B) abundances**
Data are presented in non-metric multidimensional scaling (NMDS) plots based on robust Aitchison distances. Small points represent individual samples, and large points represent group centroids at baseline and follow-up. Ellipses represent 95% confidence intervals.

## Adipose and skeletal muscle gene expression

We focused our attention on measuring the mRNA expression of genes coding for proteins reported to be important for mediating the effects of microbiota on peripheral tissues: LBP and CD14 as potential mediators of the LPS effects, and FFAR2/GPR43 and FFAR3/GPR41 as potential mediators of the effects of SCFA. Gene expression data are presented as fold change in $\Delta\Delta$Ct from baseline on a $\log^{10}$ scale to facilitate interpretation ($\Delta\Delta$Ct; Fig. 9).

In adipose tissue FFAR2/GPR43 was detected in $n = 11$ and 7 participants from control and intervention, respectively, but changes were not significantly different between groups ($P = 0.381$). FFAR3/GPR41 mRNA was detected in significantly fewer participants ($n = 4$ and 3

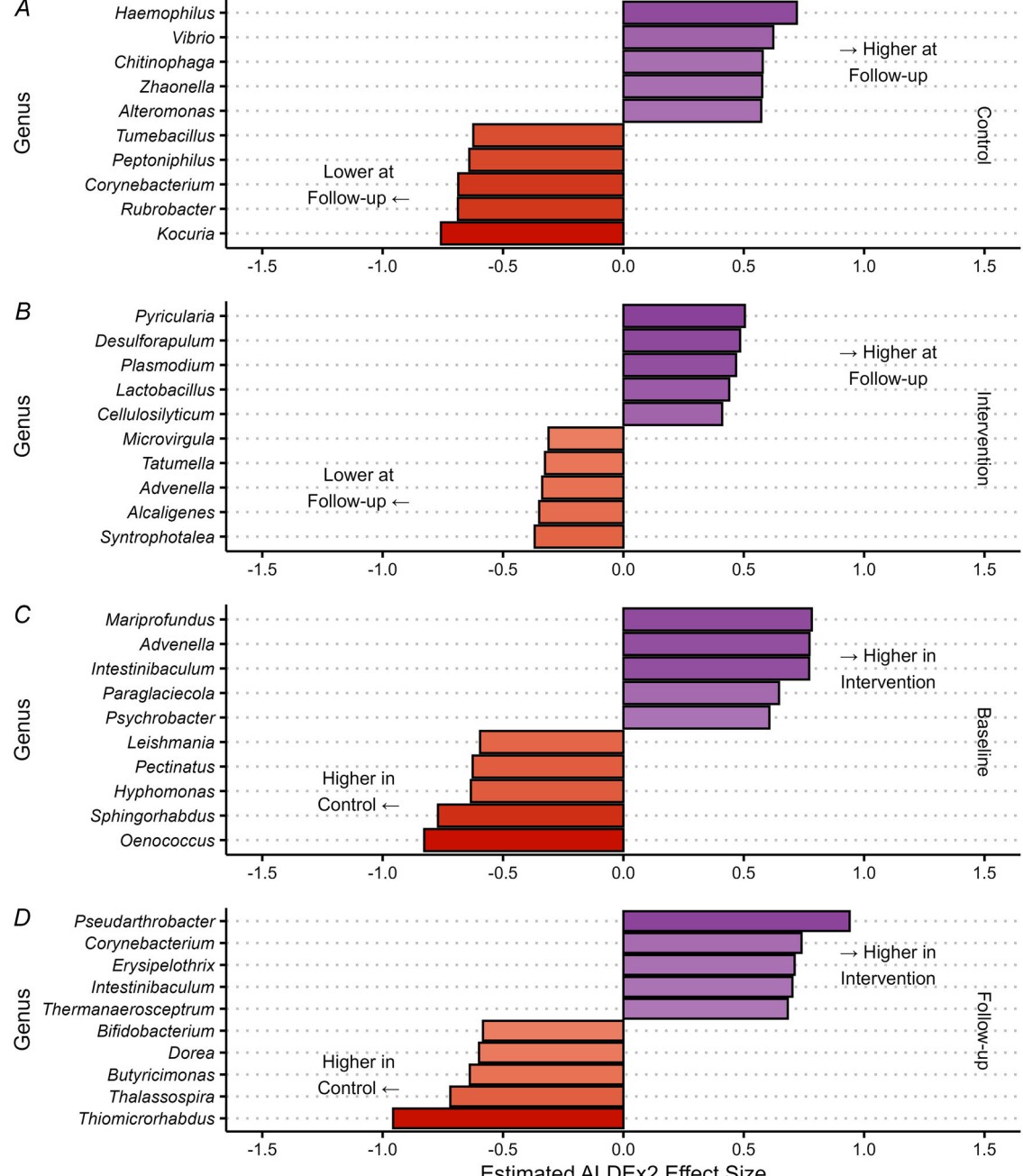

**Figure 4. Differential abundance analysis at genus level**
Largest ALDEx2 effect sizes of difference in centre log ratio transformed gut microbiome genus abundances between baseline and follow-up in control (*A*) and intervention (*B*) groups, and between groups at baseline (*C*) and follow-up (*D*). No differences were statistically significant after Benjamini Hochberg adjustment (*q* > 0.1).

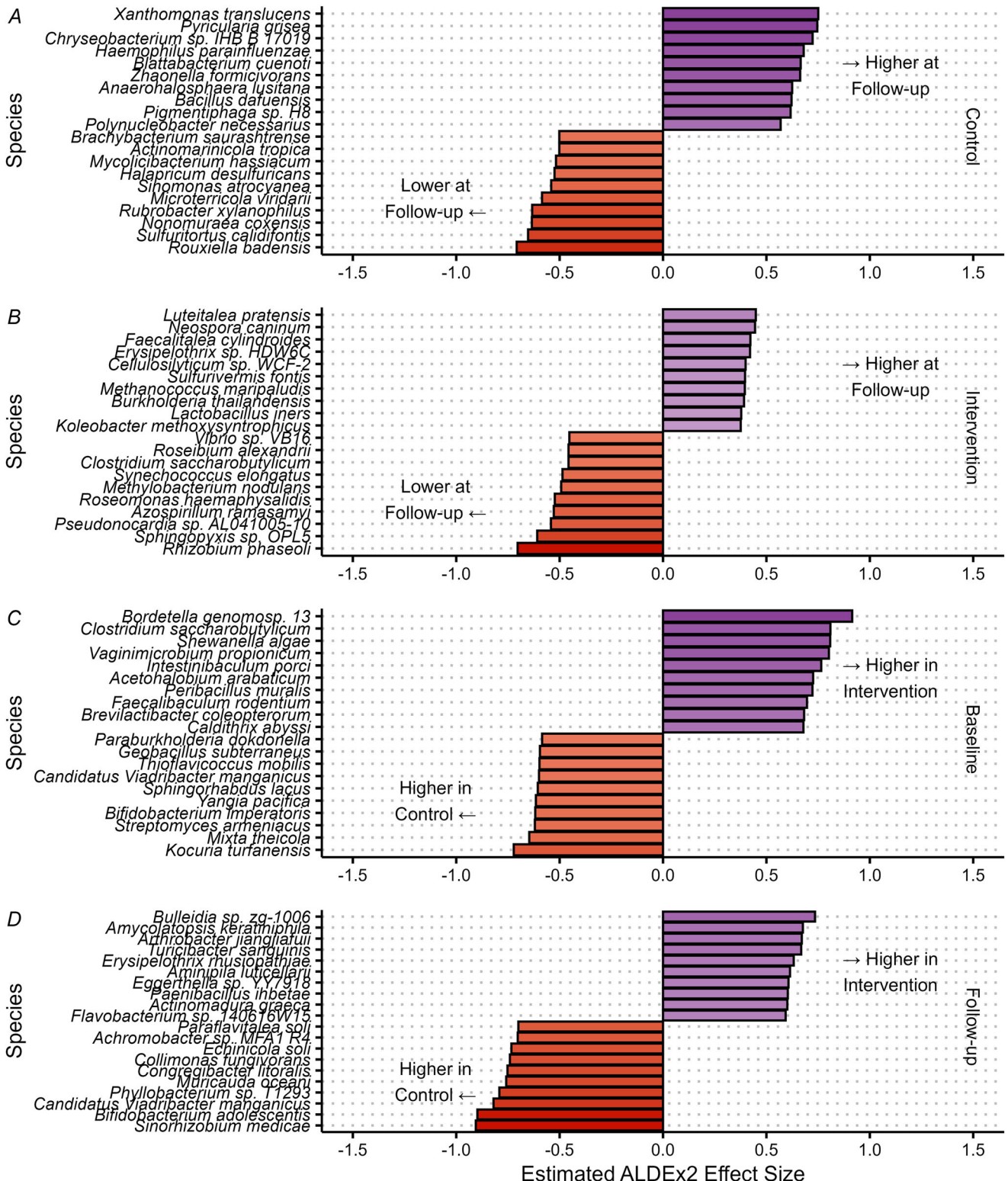

**Figure 5. Differential abundance analysis at species level**
Largest ALDEx2 effect sizes of difference in centre log ratio transformed gut microbiome species abundances between time points in control (*A*) and intervention (*B*) groups, and between groups at baseline (*C*) and follow-up (*D*). No differences were statistically significant after Benjamini Hochberg adjustment ($q > 0.1$).

participants from control and intervention, respectively). LBP and CD14 ΔΔCt values in adipose tissue were not significantly different between groups ($P = 0.606$ and $P = 0.469$, respectively; Fig. 9).

In skeletal muscle the mRNA levels of FFAR3/GPR41 were not detected (Ct values >35). Only eight individual muscle samples had detectable levels of FFAR2/GPR43, and only one pair of samples was from the same participant, which did not provide enough data to permit statistical analysis. Expression of all detected targeted genes in skeletal muscle was not significantly different between baseline and follow-up (LBP $P = 0.256$, CD14 $P = 0.529$; Fig. 9).

## Discussion

This RCT shows that the human gut microbiome remains unchanged in the face of a short-term energy restriction and vigorous-intensity exercise intervention that significantly altered body mass, body composition and clinically relevant markers of metabolic health in people with overweight/obesity.

The combination of short-term dietary energy restriction and regular vigorous-intensity exercise resulted in significant decreases in body mass, fat mass and waist circumference in the intervention group compared to control in otherwise healthy sedentary adults with over-weight/obesity. This was accompanied by significantly

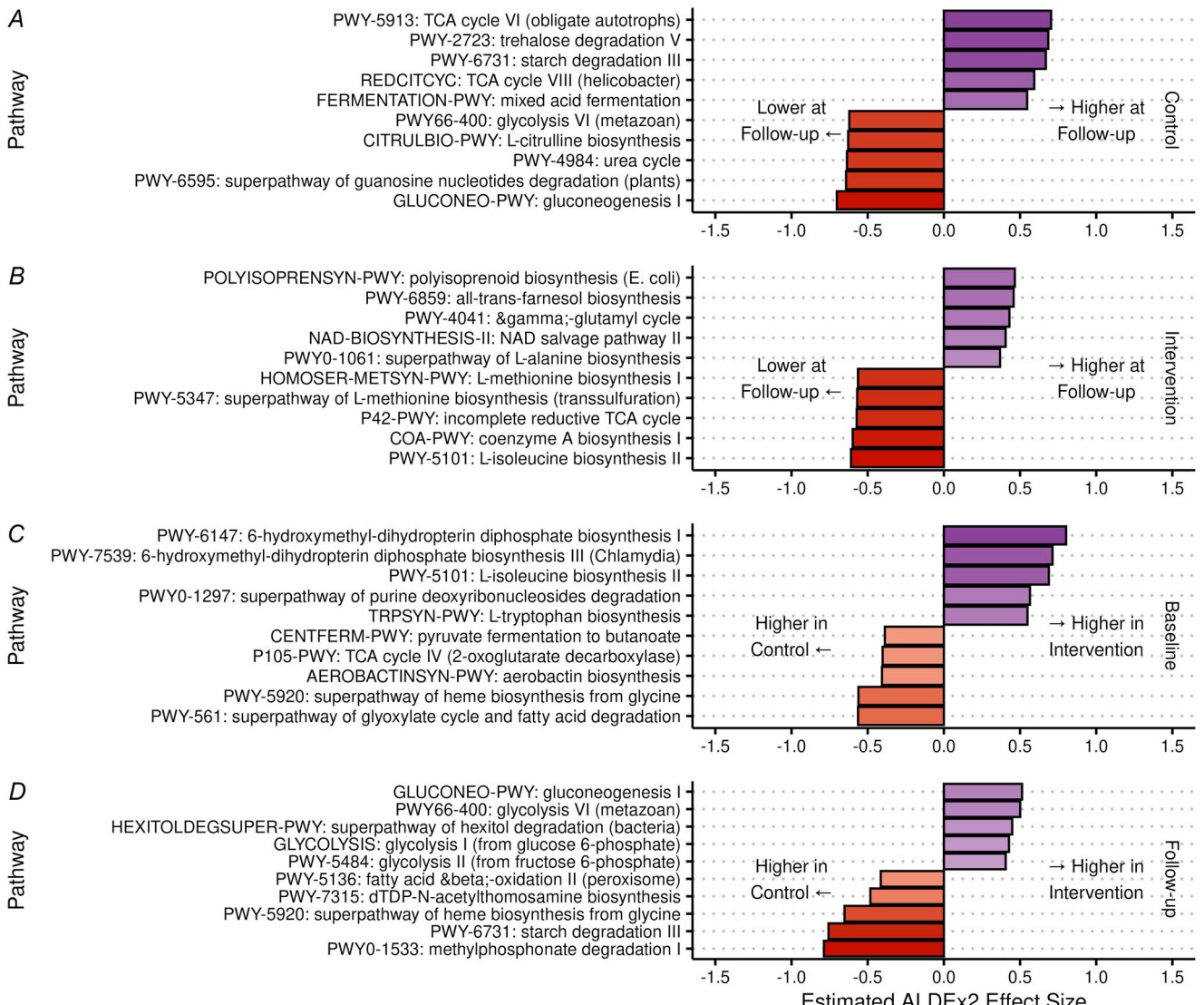

**Figure 6. Differential abundance of normalised unstratified functional pathways**
Largest ALDEx2 effect sizes of difference in centre log ratio gut microbiome normalised unstratified functional pathway abundances between time points in control (*A*) and intervention (*B*) groups, and between groups at baseline (*C*) and follow-up (*D*). No differences were statistically significant after Benjamini Hochberg adjustment (*q* > 0.1).

increased insulin sensitivity and reduced fasting insulin, total and LDL cholesterol and leptin in the intervention group. Despite these significant metabolic changes after the intervention, the composition of the gut microbiome remained stable overall, with no significant changes in Shannon index or $\beta$ diversity of species and pathway abundances. Furthermore, there was no evidence of significant microbiome changes at the functional metabolic level, including concentrations of SCFAs in serum and faeces, and gene expression of SCFA receptors and other relevant genes in subcutaneous adipose and skeletal muscle tissues. Overall these results suggest that the human gut microbiome may be robust to significant physiological changes incurred as a result of 3 weeks of dietary energy restriction and vigorous-intensity exercise in a group of sedentary men and postmenopausal women with overweight/obesity.

Several previous human intervention trials investigating the interactions between exercise and the gut microbiome have lacked a non-exercising control group, prohibiting the ability to make causal inferences about the effects of exercise (Allen et al., 2018; Cronin et al., 2018; Munukka et al., 2018). The present study included a non-exercising control group, and similar variation in the abundances of individual microbiome taxa and functional pathways was observed in this group compared to intervention. This highlights the importance of including a non-exercising control group due to the variability associated with longitudinal sampling of the gut microbiome.

Of the previous studies investigating the interactions between exercise and the human gut microbiome, three induced significant reductions in body mass and/or fat mass (Allen et al., 2018; Kern et al., 2020; Motiani et al., 2020), whereas in other interventions body and/or fat mass was either unchanged (Liu et al., 2020; Munukka et al., 2018; Quiroga et al., 2020; Rettedal et al., 2020) or not reported (Zhong et al., 2021). The RCT conducted by Kern et al. (2020) bears several similarities to the current work. Both studies used similar-sized groups of participants with overweight/obesity who were otherwise healthy, and both induced a significant decrease in fat mass through vigorous-intensity exercise compared to a non-exercising control group. In contrast to the current study, Kern et al. (2020) observed significant increases in Shannon index in response to 3 months of vigorous-intensity exercise training compared to the 3-week intervention used in our trial. Importantly the diet of participants in the study by Kern et al. (2020) was deliberately not standardised as part of the 'free-living' nature of the study. Because diet is known to drive changes in gut microbiome taxonomic composition and functional potential, especially in the context of people who exercise compared to those who do not (Barton et al., 2018; Bressa et al., 2017; Clarke et al., 2014), it is possible that the changes observed by Kern et al. (2020) stem from changes to dietary composition during the intervention period. However, other non-dietary differences between the present study and Kern and colleagues (Kern et al., 2020) could explain the divergent findings, including differences in study time scale, sequencing methodology and differences in

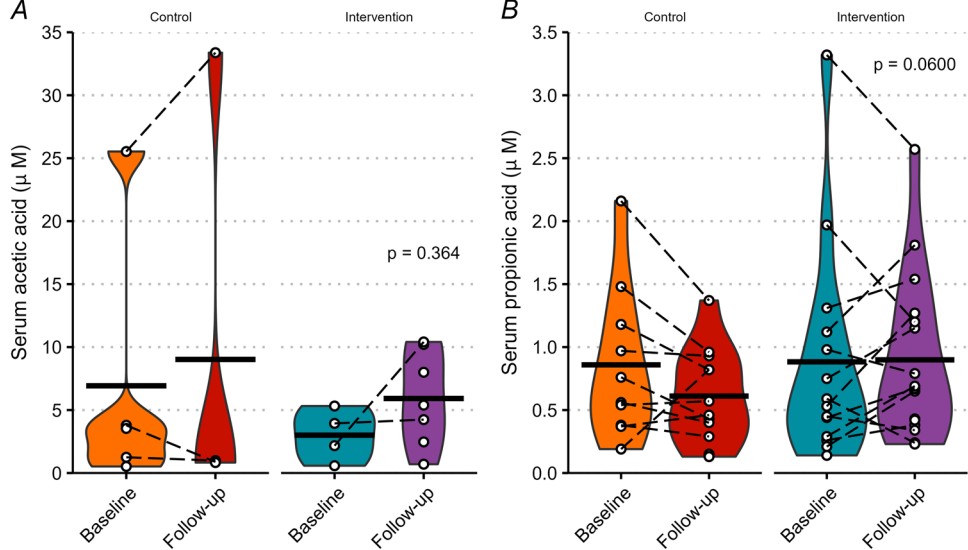

**Figure 7. Serum short-chain fatty acid concentrations**
Serum acetic acid (*A*) and propionic acid (*B*) concentrations. Black crossbars represent the means. White circles represent individual data points, joined by dashed lines to represent the same participant. *P*-values show ANCOVA (analysis of covariance) effect of group. Missing values are due to some samples being below the limit of quantitation.

statistical approaches. It is also important to highlight that Kern et al. did not investigate the impact of the intervention on host metabolism and/or health biomarkers (Kern et al., 2020). In the present study by investigating the interactions between exercise, energy restriction and gut microbiome over a carefully controlled shorter time period, and using a protocol that has established effects on metabolism (Walhin et al., 2016), we have provided novel evidence that metabolically important changes occur in this population prior to and/or without a change in gut microbiome composition. This does not discount the potential importance of the microbiome for metabolic health in the longer term, but it does indicate that changes to the microbiome are probably not required for rapid improvements to metabolic health with short-term weight loss.

SCFAs have been implicated as one of the links between the gut microbiome and metabolic health (Canfora et al.,

2015, 2019). Allen et al. (2018) compared the effects of 6 weeks of exercise on the gut microbiome of lean and obese adults without a non-exercising control group and observed a shift in $\beta$ diversity whereby the taxonomic composition of the lean and obese groups was significantly different at baseline but not significantly different at post-exercise training follow-up. In addition there was an increase in the concentrations of faecal acetate and butyrate after exercise only in the lean group, whereas faecal propionate somewhat increased in both groups, although this was largely driven by increases in the lean group. In the current study we observed that 3 weeks of exercise training did not significantly affect faecal concentrations of these three SCFAs in participants with overweight/obesity, which concurs with the findings of Allen et al. (2018) for their participants with obesity. Thus, based on the collective findings from previous studies using pre-post comparisons (Allen et al., 2018), and the

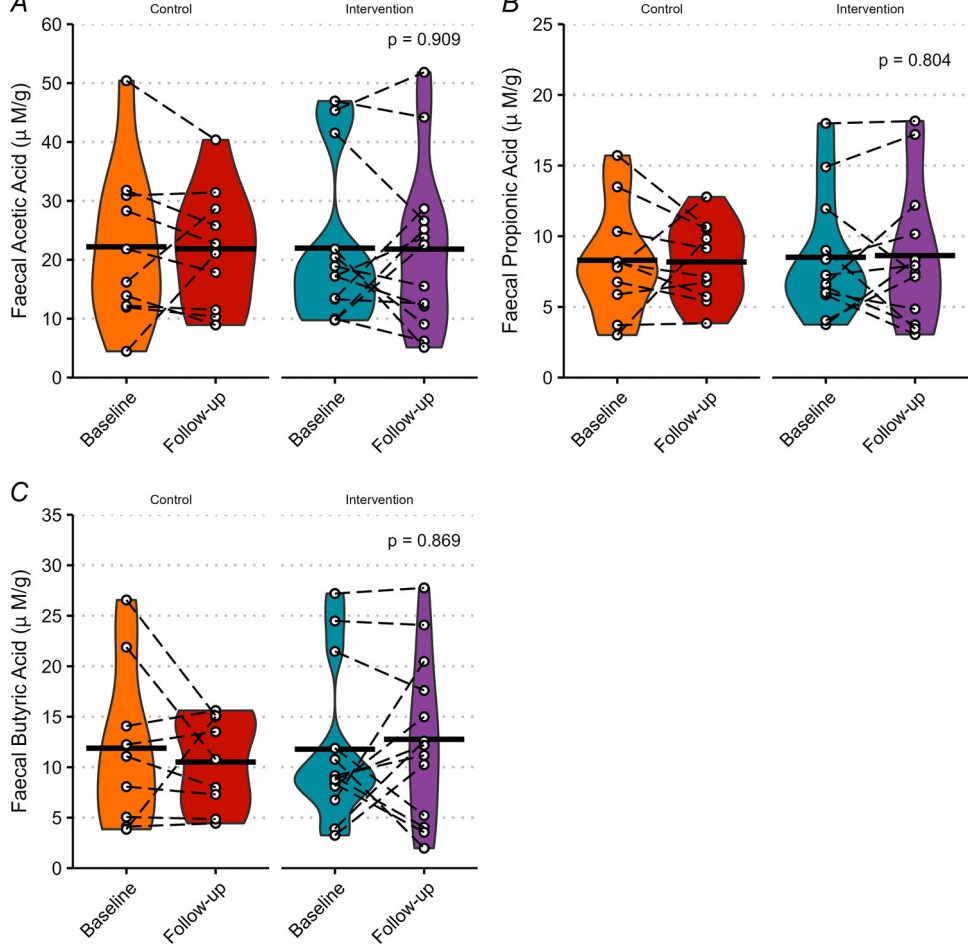

**Figure 8. Faecal short-chain fatty acid concentrations**
Faecal acetic acid (*A*), propionic acid (*B*) and butyric acid (*C*) concentrations. Black crossbars represent the means. White circles represent individual data points, joined by dashed lines to represent the same participant. *P*-values show ANCOVA (analysis of covariance) effect of group. Missing values are due to some samples being below the limit of quantitation.

present RCT, it appears as though exercise interventions with or without weight loss do not change faecal SCFAs or fasting serum SCFA concentrations in people with overweight/obesity, at least in the short term (3–6 weeks).

Another route through which metabolic health may be affected via the gut microbiome is translocation of LPS from the gut lumen and the resultant stimulation of an inflammatory response. We assessed LBP given the difficulties associated with measuring LPS directly. We also assessed the concentration of CRP and the expression of inflammatory genes associated with metabolic endotoxaemia. Despite improvements in metabolic health markers, we observed no significant differences in CRP or LBP, and no significant effects on LBP and CD14 gene expression after this short-term intervention.

We measured the expression of the receptors for SCFAs in two metabolically important tissues (subcutaneous adipose and skeletal muscle) for the first time in a human intervention study investigating the inter-

actions between exercise, weight loss and gut microbiome. SCFAs are thought to influence metabolic health via these receptors (Al Mahri et al., 2022; Frampton et al., 2020). This study adds valuable human data regarding the expression of these genes and confirms that both FFAR2/GPR43 and FFAR3/GPR41 genes are expressed in human subcutaneous adipose tissue, although not at detectable levels in all individuals. Although we did observe numerical increases in the levels of FFAR2/GPR43 and FFAR3/GPR41 mRNA in adipose tissue in response to the intervention compared to control, these changes were not statistically significant and highly variable. Further studies are needed to determine whether SCFA receptors in adipose and muscle respond to exercise and diet interventions.

It has been observed that acute bouts of exercise can potentially influence the composition of the human gut microbiome and the concentrations of metabolites in faeces and serum (Tabone et al., 2021). It is not clear

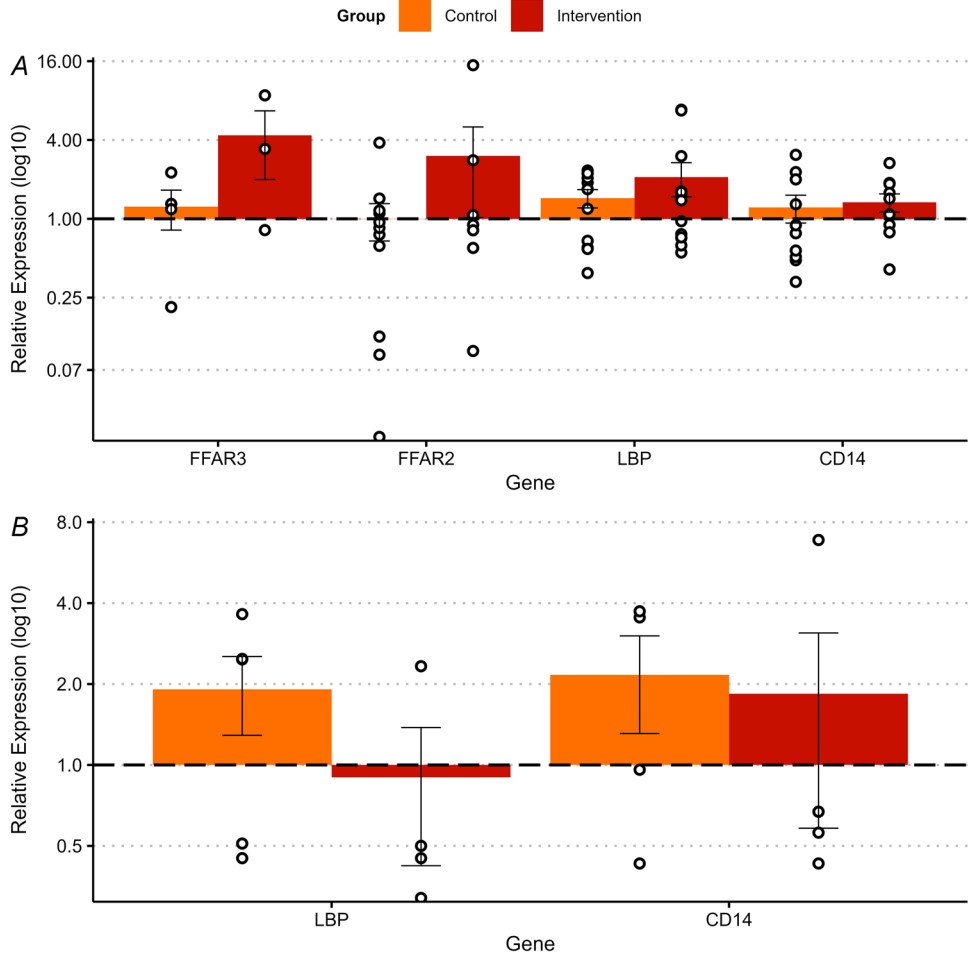

**Figure 9. Gene expression in subcutaneous adipose (A) and skeletal muscle (B) tissues**
Data are normalised to housekeeper (PPIA (peptidylprolyl isomerase A), GAPDH and $\beta$-Actin) and shown as mean fold change in $\Delta\Delta Ct$ from baseline $\pm$ SEM $\Delta\Delta Ct$ on a log$_{10}$scale. Samples outside the detectable limit (Ct > 35) were excluded from the analysis.

whether there was a defined period without exercise prior to follow-up in the RCT by Kern et al. (2020), whereas at least 36 h separated the last exercise bout from the follow-up measurements in the present study. Thus, there could be short-term changes associated with each exercise bout which may need to be considered as part of the overall training stimulus (Thompson et al., 2012). It is also unclear whether episodic postprandial changes in SCFA concentration could play a role in host metabolism (Meiller et al., 2023), and these will not be reflected in the fasting measurements included in the present study. There is a need for further research to examine temporal responses to exercise and feeding, including interactions between acute and chronic responses. We should also highlight that hydration and stool frequency were not measured during the study period, which may affect gut microbiome measures. Furthermore, the sample size calculation for this study was based on changes in fasting insulin, and it is possible that changes to the microbiome and microbiome-related outcomes (e.g. SCFAs, LBP, gene expression) are more variable and require a different sample size. However there was no indication that any of these parameters would have been significantly different in the present study with a larger sample size.

In conclusion, short-term vigorous-intensity exercise and energy restriction significantly improved several markers of metabolic health without modifying the gut microbiome, concentrations of SCFAs in faeces and serum or the expression of the SCFA receptors in skeletal muscle or subcutaneous adipose tissue. The possibility remains that the gut microbiome may play an important role in regulating metabolic health in the longer term, but these data suggest that changes to $\alpha$ and $\beta$ diversity and relative abundance may not be required to induce early metabolic health improvements from vigorous-intensity exercise and energy restriction.

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

## Additional information

### Data availability statement

The anthropometric, physiological and biochemical data for this study are available at the University of Bath Research Data Archive: https://doi.org/10.15125/BATH-01463.

Shotgun metagenomics data for this study are available at the European Nucleotide Archive https://www.ebi.ac.uk/ena/browser/ (data accession ID: PRJEB836868).

For the purpose of Open Access, the author has applied a Creative Commons Attribution (CC BY) licence to any Author Accepted Manuscript version arising.

### Competing interests

P.D.C. has been funded by PrecisionBiotics Group, Friesland Campina, Danone and PepsiCo. P.D.C. has also received funding to travel to or present at meetings from H&H, the National Dairy Council U.S., PepsiCo, Abbott, Arla and Yakult. In addition P.D.C. is the co-founder and CTO of SeqBiome Ltd, a provider of sequencing and bioinformatics services for microbiome analysis. J.A.B. is an investigator on research grants funded by BBSRC, MRC, NIHR, the British Heart Foundation, the Rare Disease Foundation, the EU Hydration Institute, GlaxoSmithKline, Nestlé, Lucozade Ribena Suntory, ARLA Foods, Cosun Nutrition Center, the American Academy of Sleep Medicine Foundation, Salus Optima (L3M Technologies Ltd) and the Restricted Growth Association; has completed paid consultancy for PepsiCo, Kellogg's, SVGC and Salus Optima (L3M Technologies Ltd); is company director of Metabolic Solutions Ltd; receives an annual honorarium as a member of the academic advisory board for the International Olympic Committee Diploma in Sports Nutrition; and receives an annual stipend as editor-in chief of the *International Journal of Sport Nutrition & Exercise Metabolism*. All other authors have no conflicts of interest to declare.

## Author contributions

Conceptualisation: R.G.D., L.A.W., J.-P.W., J.T.G., J.A.B., F.K. and D.T.; methodology: R.G.D., L.A.W., A.H., C.O., W.B., F.C., J.-P.W., M.A.V.-G., I.G.-P., G.F., O.O., P.D.C., J.T.G., J.A.B., F.K. and D.T.; investigation: R.G.D., L.A.W., A.H., J.-P.W., J.T.G., J.A.B., F.K. and D.T.; analysis: R.G.D., C.O., W.B., F.C., M.A.V.-G., I.G.-P., F.K. and D.T.; writing of the first draft: R.G.D., F.K. and DT; writing, interpretation, reviewing and editing: all co-authors. All authors approved the final version. All authors agree to be accountable for all aspects of the work in ensuring that questions related to the accuracy or integrity of any part of the work are appropriately investigated and resolved. All persons designated as authors qualify for authorship, and all those who qualify for authorship are listed.

## Funding

UKRI | Medical Research Council (MRC): Dylan Thompson, MR/P002927/1; UKRI | Medical Research Council (MRC): Javier T Gonzalez, MR/P002927/1; UKRI | Medical Research Council (MRC): James A Betts, MR/P002927/1; UKRI | Medical Research Council (MRC): Francoise Koumanov, MR/P002927/1

## Acknowledgements

This research was supported by funding from the Medical Research Council (MR/P002927/1), the University of Bath and Ian Tarr.

## Keywords

metabolism, microbiome, weight loss

## Supporting information

Additional supporting information can be found online in the Supporting Information section at the end of the HTML view of the article. Supporting information files available:

**Peer Review History**

