## [Peer Review History · The Journal of Physiology]

Effects of a combined energy restriction and vigorous intensity exercise intervention on the human gut microbiome - a randomised controlled trial

Dylan Thompson, Russell Graham Davies, Laura A Wood, Ciara O'Donovan, Wiley Barton, Fiona Crispie, Jean-Philippe Walhin, Maria A Valdivia-Garcia, Gary Frost, Orla O'Sullivan, Paul D Cotter, Javier T Gonzalez, James A Betts, Francoise Koumanov, and Aaron Hengist

DOI: 10.1113/JP287424

Corresponding author(s): Dylan Thompson (D.Thompson@bath.ac.uk)

The following individual(s) involved in review of this submission have agreed to reveal their identity: Jamie Pugh (Referee #1)

Review Timeline:

Submission Date:	29-Sep-2024
Editorial Decision:	12-Feb-2025
Revision Received:	03-May-2025
Accepted:	09-Jul-2025

Senior Editor: Kim Barrett

Reviewing Editor: Stephen Keely

Transaction Report:

Dear Dr Thompson,

Re: JP-RP-2024-287424 "Effects of a combined energy restriction and vigorous intensity exercise intervention on the human gut microbiome - a randomised controlled trial" by Dylan Thompson, Russell Graham Davies, Laura A Wood, Ciara O'Donovan, Wiley Barton, Fiona Crispie, Jean-Philippe Walhin, Maria A Valdivia-Garcia, Gary Frost, Orla O'Sullivan, Paul D Cotter, Javier T Gonzalez, James A Betts, and Francoise Koumanov

Please accept our sincere apologies for the delay in providing you with an editorial decision on your submission. We have appreciated your patience while we waited for review reports to be submitted.

Thank you for submitting your manuscript to The Journal of Physiology. It has been assessed by a Reviewing Editor and by 3 expert referees and we are pleased to tell you that it is potentially acceptable for publication following satisfactory major revision.

Please address all the points raised and incorporate all requested revisions or explain in your Response to Referees why a change has not been made. We hope you will find the comments helpful and that you will be able to return your revised manuscript within 2 months. If you require longer than this, please contact journal staff: jp@physoc.org. Please note that this letter does not constitute a guarantee for acceptance of your revised manuscript.

REVISION CHECKLIST:

We look forward to receiving your revised submission.

Yours sincerely,

Kim Barrett
Senior Editor
The Journal of Physiology

REQUIRED ITEMS

- Author photo and profile. First or joint first authors are asked to provide a short biography (no more than 100 words for one author or 150 words in total for joint first authors) and a portrait photograph. These should be uploaded and clearly labelled together in a Word document with the revised version of the manuscript. See Information for Authors for further details.

- Please include an Abstract Figure file, as well as the Figure Legend text within the main article file. The Abstract Figure is a piece of artwork designed to give readers an immediate understanding of the research and should summarise the main conclusions. If possible, the image should be easily 'readable' from left to right or top to bottom. It should show the physiological relevance of the manuscript so readers can assess the importance and content of its findings. Abstract Figures should not merely recapitulate other figures in the manuscript. Please try to keep the diagram as simple as possible and without superfluous information that may distract from the main conclusion(s). Abstract Figures must be provided by authors no later than the revised manuscript stage and should be uploaded as a separate file during online submission labelled as File Type 'Abstract Figure'. Please also ensure that you include the figure legend in the main article file. All Abstract Figures should be created using BioRender. Authors should use The Journal's premium BioRender account to export high-resolution images. Details on how to use and access the premium account are included as part of this email.

EDITOR COMMENTS

Reviewing Editor:

This is an interesting and well-conducted and presented study. All 3 Reviewers note numerous strengths and merits to the study and note that it could have important implications for how we understand the role of the microbiota in mediating the health benefits of dietary and exercise interventions. However, all 3 Reviewers also raised significant concerns regarding the limitations of the study and interpretation of the results. In particular this relates to concerns with respect to the appropriateness of the study design, adequate powering, rigour of the microbiome analysis, and a lack of information regarding dietary restrictions. The authors would need to fully address these concerns if resubmitting.

Please also see 'Required Items' above.

Senior Editor:

Please accept our very sincere apologies for the delays that occurred during the review of your work. One of the reviewers experienced a significant family emergency that impeded his/her workload and communication for a considerable period. With the benefit of hindsight, we likely should have cut our losses earlier, but we believed that the reviewer had special

expertise to share. We will make every effort to ensure that any revised manuscript is acted upon much more promptly.

REFeree COMMENTS

Referee #1:

I would like to thank the authors for their hard work and efforts in putting this together. It is clear a lot of work has gone into the planning and writing of this work. Please find my comments below. I'm always conscious of how comments can come across or be misconstrued. If the writing appears short and direct, this is to be concise, not to be rude. Equally, all comments below are written with the intention to challenge the writing and create a better manuscript, not to offend the authors.

Please find comments below.

While this is a very well designed and controlled study, which I think will have a high impact in this area of study, there are a couple of concerns that I believe require further thought and discussion.

Introduction

The introduction is very well written, although the two paragraphs between line 115-146 feel longer than necessary. These sections contain a lot of repeated citations and could be made more concise. Especially given that no mention of any specific microbiome composition changes expected or seen previously, other than diversity.

Line 110 - von Schwartzberg et al. 2021 link the changes in microbiome to the severe reduction in calories, not necessarily the weight loss. This is supported by many of the measures returning to baseline values once calorie intake was increased, despite body mass remained lower than baseline. I suggest removing and leaving only the systematic review from Koutoukidis et al.

Line 115-127 - I think it should be clarified, that this is referring to long-term exercise adaptations, as opposed to acute exercise. Particularly as there are a number of studies that have investigated the acute effects of exercise on SCFA (see PMID: 37702557)

Methods

Line 183 - why was 3 months selected for last potential antibiotic treatment? There is evidence of sustained effects even after 6 (PMID: 35417701) and 24 months (PMID: 18043614). Was any previous antibiotic use recorded?

Line 143 - should the limitations of dietary intake measures be reported? Participant characteristics already show an average deficit of 400-500 kcal/day for each group at baseline, which is likely to be due to underestimation of energy intake.

Line 200 - why was the power calculation driven by changes in fasting plasma, if the aim was to investigate differences in the microbiome? Was this sufficiently powered?

Line 356 - were there any dietary controls prior to stool sample collection?

Line 360 - given the manuscript title and aims, why were these participants not excluded from the overall analyses given that this sample is central to the study design?

Line 365 - what was the time until processing and storing at -80?

Line 488 - Why was the Benjamini-Hochberg method used? While it has been used in studies with larger sample sizes, it has been suggested to be too conservative and likely the reason for studies finding few or no differences between groups (PMID: 30045760)

Table 2 - there may be some rounding errors but some of the 'Change' values are different to the calculated Pre - Post differences reported.

Discussion

Line 577 - suggest removing the descriptors 'intensive' and 'vigorous'

Line 582 - as above

Line 621 - While Kern et al., did not control for diet, the present study only recorded participant self-reported adherence to their typical diet. The differences in outcome could also be due to differences in the timescale, 16s vs shotgun sequencing, or differences in the statistical approach to the data.

Referee #2:

The study by Thompson et al aims to examine early changes to the microbiome in overweight and obese individuals, using calorie restriction and increased activity as an intervention and compare to a non-intervention group. The authors examine the microbiota via shotgun metagenomics and correlate with tissue and blood markers of energy metabolism as well as physiological indices. Using alpha and beta diversity as the primary measure, they conclude that early weight loss and metabolism is unlikely to be mediated by the gut microbiome.

The manuscript is well written and the study is generally well presented. There are some concerns with the approach and design of the study, which may not be sufficient to reach the authors conclusions.

I have provided comments below which the authors may wish to consider:

1. The major critique of the study is that the approach to study influence of diet/exercise on the microbiota is rather descriptive and not well defined. For instance, it appears that dietary restriction was approached in a very heterogeneous manner, with self-reported and self-managed dietary intervention. This in itself is prone to inaccuracy, but the major issue lies in the lack of data on how the dietary restriction was achieved. One participant may have, for instance, focused on significant reduction of fats, while another focused on carbohydrates while another may have reduced portion sizes. Even

then, the habitual diet may vary widely between participants making the impact of these changes difficult to interpret. Without this specific dietary data, it is impossible to uniformly imply that the dietary intervention does or does not impact the microbiota's ecology in a manner that is less obvious than alpha/beta diversity but significant none the less.

2. With this in mind, the authors should provide details on the participants habitual intake and some measurement of participant food intake during the intervention would be advantageous, even a basic breakdown of protein, fats, carbohydrates and fibre intake is likely to be informative, especially when examining changes to the microbiota at a metagenomic level.

3. Was hydration/fluid intake over the intervention period measured? Similarly was there any attempt to measure motility/stool frequency? These may impact the study endpoints.

4. A statement on exercise adherence was provided, how did the authors measure diet adherence?

5. The second major critique pertains to the study power. The power calculation is based on detecting a reduction in fasting plasma insulin. While this would potentially provide sufficient power for metabolic analyses, it is unclear how this measurement justifies powering the more complex microbiome analyses. Even with that, the study did not analyse data from the required number of participants for the microbiome data (N=23, 9 control, 14 intervention analysed, vs N=24, 8 controls, 16 interventions required). It would be useful if the authors provided an explanation of why the samples for each analysis varies so widely. Figure 1 shows 18 vs 12 for analysis, but this is somewhat misleading for the major endpoints discussed.

6. With respect to point 1 and the microbial ecology, have the authors considered analysis of the microbiome at the functional level? The microbial genes related to dietary breakdown and energy harvesting could be assessed and, correcting for the sample drop outs, up/downstream pathways related to the metabolic measurements could be investigated. This would lessen the descriptive nature of the work.

7. The discussion should provide a paragraph on the limitations of the study, particularly pertaining to power.

8. It is unclear why the authors did not include individual intervention groups (diet or exercise) in the experimental design, as the current design would make it impossible to ascertain whether one intervention dominates or is even required.

Referee #3:

Summary:

This study investigated whether a 3-week combined exercise and dietary energy restriction intervention in sedentary adults with overweight/obesity leads to changes in the gut microbiome, potentially mediating improvements in metabolic health. Despite significant reductions in body mass, fat mass, and improvements in insulin sensitivity, no significant changes in gut microbiome diversity or composition were observed. The study is methodologically sound and addresses a relevant question, though the interpretation of results and presentation of microbiome data would benefit from some revisions to strengthen the manuscript's impact and clarity.

Strengths:

1. Robust Study Design: The randomized controlled trial (RCT) design strengthens causal inference, addressing a critical gap in existing literature that mostly comprises observational studies.
2. Comprehensive Methodology: Use of shotgun metagenomics (Kraken2/Bracken and HUMAnN2) provided in-depth analysis of gut microbiome changes.
3. Multi-Tissue Analysis: The study uniquely measured SCFA concentrations and gene expression related to host-microbiome interactions in both skeletal muscle and adipose tissue.
4. Clear Presentation of Results: The manuscript presents results concisely with appropriate statistical analyses, and figures/tables are well-structured.

Weaknesses and Areas for Improvement:

1. Short Intervention Duration:

o A 3-week period may be insufficient to observe gut microbiome changes. Longer interventions could reveal delayed microbial adaptations.

o Recommendation: Discuss limitations of the short duration and suggest future studies with extended timelines. Include the acute nature of the intervention in statements such as "...the human gut microbiome remains stable in the face of an intensive energy restriction and vigorous exercise intervention".

2. Sample Size and Power Calculation:

o Although justified, the modest sample size (n=30) may limit detection of subtle microbiome shifts.

o Recommendation: Include a discussion on whether the study was sufficiently powered to detect microbiome changes.

3. Dietary Composition Control:

o The study maintained participants' habitual diets but only reduced energy intake. Specific dietary components influencing the microbiome (e.g., fiber, prebiotics) were not controlled.

o Recommendation: Discuss how unaltered diet composition might limit microbiome responsiveness.

4. Microbiome Data Presentation:

o Alpha and beta diversity metrics are reported, but taxonomic and functional pathway data are underexplored.

o Recommendation: Consider providing more detailed analyses of specific taxa or pathways potentially affected by the intervention.

5. Interpretation of Null Findings:

o The conclusion that early metabolic improvements are independent of microbiome changes is strong but may overlook other microbial functions not captured by the methods used.

o Recommendation: Use hedging language (e.g., "may not be mediated") and discuss alternative mechanisms. Furthermore, include the acute nature of the study in the language that "metabolic improvements are independent of microbiome changes".

Minor Issues:

- Grammar and Clarity: Minor grammatical improvements could enhance clarity, particularly in the Discussion section.
- Figure Legends: Ensure all figures and tables are fully self-explanatory with clear legends.

END OF COMMENTS

Summary:

This study investigated whether a 3-week combined exercise and dietary energy restriction intervention in sedentary adults with overweight/obesity leads to changes in the gut microbiome, potentially mediating improvements in metabolic health. Despite significant reductions in body mass, fat mass, and improvements in insulin sensitivity, no significant changes in gut microbiome diversity or composition were observed. The study is methodologically sound and addresses a relevant question, though the interpretation of results and presentation of microbiome data would benefit from some revisions to strengthen the manuscript's impact and clarity.

Strengths:

1. **Robust Study Design:** The randomized controlled trial (RCT) design strengthens causal inference, addressing a critical gap in existing literature that mostly comprises observational studies.
2. **Comprehensive Methodology:** Use of shotgun metagenomics (Kraken2/Bracken and HUMAnN2) provided in-depth analysis of gut microbiome changes.
3. **Multi-Tissue Analysis:** The study uniquely measured SCFA concentrations and gene expression related to host-microbiome interactions in both skeletal muscle and adipose tissue.
4. **Clear Presentation of Results:** The manuscript presents results concisely with appropriate statistical analyses, and figures/tables are well-structured.

Weaknesses and Areas for Improvement:

1. **Short Intervention Duration:**
 - A 3-week period may be insufficient to observe gut microbiome changes. Longer interventions could reveal delayed microbial adaptations.
 - *Recommendation:* Discuss limitations of the short duration and suggest future studies with extended timelines. Include the acute nature of the intervention in statements such as "...the human gut microbiome remains stable in the face of an intensive energy restriction and vigorous exercise intervention".
2. **Sample Size and Power Calculation:**
 - Although justified, the modest sample size (n=30) may limit detection of subtle microbiome shifts.
 - *Recommendation:* Include a discussion on whether the study was sufficiently powered to detect microbiome changes.
3. **Dietary Composition Control:**
 - The study maintained participants' habitual diets but only reduced energy intake. Specific dietary components influencing the microbiome (e.g., fiber, prebiotics) were not controlled.
 - *Recommendation:* Discuss how unaltered diet composition might limit microbiome responsiveness.
4. **Microbiome Data Presentation:**
 - Alpha and beta diversity metrics are reported, but taxonomic and functional pathway data are underexplored.

- *Recommendation:* Consider providing more detailed analyses of specific taxa or pathways potentially affected by the intervention.
5. **Interpretation of Null Findings:**
- The conclusion that early metabolic improvements are independent of microbiome changes is strong but may overlook other microbial functions not captured by the methods used.
 - *Recommendation:* Use hedging language (e.g., "may not be mediated") and discuss alternative mechanisms. Furthermore, include the acute nature of the study in the language that “metabolic improvements are independent of microbiome changes”.

Minor Issues:

- **Grammar and Clarity:** Minor grammatical improvements could enhance clarity, particularly in the Discussion section.
- **Figure Legends:** Ensure all figures and tables are fully self-explanatory with clear legends.

To the ed:

JP-RP-2024-287424 "Effects of a combined energy restriction and vigorous intensity exercise intervention on the human gut microbiome - a randomised controlled trial" by Dylan Thompson, Russell Graham Davies, Laura A Wood, Ciara O'Donovan, Wiley Barton, Fiona Crispie, Jean-Philippe Walhin, Maria A Valdivia-Garcia, Gary Frost, Orla O'Sullivan, Paul D Cotter, Javier T Gonzalez, James A Betts, and Francoise Koumanov

We would like to thank the reviewers and editors for their helpful comments on our work. We have responded to each comment in blue font under each comment. We have used track changes in the associated manuscript file.

As discussed with the editorial office just after initial submission, we unfortunately omitted one author from the initial submission. As discussed, we have now completed the required paperwork to add this author (Aaron Hengist). Apologies for the original omission.

REQUIRED ITEMS

- Author photo and profile. First or joint first authors are asked to provide a short biography (no more than 100 words for one author or 150 words in total for joint first authors) and a portrait photograph. These should be uploaded and clearly labelled together in a Word document with the revised version of the manuscript. See Information for Authors for further details.
- Please include an Abstract Figure file, as well as the Figure Legend text within the main article file. The Abstract Figure is a piece of artwork designed to give readers an immediate understanding of the research and should summarise the main conclusions. If possible, the image should be easily 'readable' from left to right or top to bottom. It should show the physiological relevance of the manuscript so readers can assess the importance and content of its findings. Abstract Figures should not merely recapitulate other figures in the manuscript. Please try to keep the diagram as simple as possible and without superfluous information that may distract from the main conclusion(s). Abstract Figures must be provided by authors no later than the revised manuscript stage and should be uploaded as a separate file during online submission labelled as File Type 'Abstract Figure'. Please also ensure that you include the figure legend in the main article file. All Abstract Figures should be created using BioRender. Authors should use The Journal's premium BioRender account to export high-resolution images. Details on how to use and access the premium account are included as part of this email.

Please find enclosed an author photo and profile. Please also find an abstract figure file. The graphical abstract legend has been added to the manuscript file.

EDITOR COMMENTS

Reviewing Editor:

This is an interesting and well-conducted and presented study. All 3 Reviewers note numerous strengths and merits to the study and note that it could have important implications for how we understand the role of the microbiota in mediating the health benefits of dietary and exercise interventions. However, all 3 Reviewers also raised significant

concerns regarding the limitations of the study and interpretation of the results. In particular this relates to concerns with respect to the appropriateness of the study design, adequate powering, rigour of the microbiome analysis, and a lack of information regarding dietary restrictions. The authors would need to fully address these concerns if resubmitting.

Please also see 'Required Items' above.

We have addressed reviewers' concerns in our response and made appropriate changes to the manuscript file.

Senior Editor:

Please accept our very sincere apologies for the delays that occurred during the review of your work. One of the reviewers experienced a significant family emergency that impeded his/her workload and communication for a considerable period. With the benefit of hindsight, we likely should have cut our losses earlier, but we believed that the reviewer had special expertise to share. We will make every effort to ensure that any revised manuscript is acted upon much more promptly.

We understand the difficulties faced by reviewers and we appreciate them taking the time to provide comments on our work.

REFeree COMMENTS

Referee #1:

I would like to thank the authors for their hard work and efforts in putting this together. It is clear a lot of work has gone into the planning and writing of this work. Please find my comments below. I'm always conscious of how comments can come across or be misconstrued. If the writing appears short and direct, this is to be concise, not to be rude. Equally, all comments below are written with the intention to challenge the writing and create a better manuscript, not to offend the authors.

We would like to thank the referee for taking the time to review our manuscript and for their kind words regarding the quality of the work. We take all comments on board as constructive criticism, and we are grateful for the opportunity to submit a revised manuscript for consideration.

Please find comments below.

While this is a very well designed and controlled study, which I think will have a high impact in this area of study, there are a couple of concerns that I believe require further thought and Discussion.

Introduction

The introduction is very well written, although the two paragraphs between line 115-146 feel longer than necessary. These sections contain a lot of repeated citations and could be made more concise. Especially given that no mention of any specific microbiome composition changes expected or seen previously, other than diversity.

We agree that the original text was too long. We have merged two paragraphs to highlight the key points.

Line 110 - von Schwartzenberg et al. 2021 link the changes in microbiome to the severe

reduction in calories, not necessarily the weight loss. This is supported by many of the measures returning to baseline values once calorie intake was increased, despite body mass remained lower than baseline. I suggest removing and leaving only the systematic review from Koutoukidis et al.

We have made the suggested change.

Line 115-127 - I think it should be clarified, that this is referring to long-term exercise adaptations, as opposed to acute exercise. Particularly as there are a number of studies that have investigated the acute effects of exercise on SCFA (see PMID: 37702557)

This point has been addressed as part of the response to the previous comment, and we have added the word 'regular' to highlight that we are referring to repeated exercise not a single bout.

Methods

Line 183 - why was 3 months selected for last potential antibiotic treatment? There is evidence of sustained effects even after 6 (PMID: 35417701) and 24 months (PMID: 18043614). Was any previous antibiotic use recorded?

Antibiotic use is common, and 3 months was selected as a reasonable compromise to allow time for the microbiome to normalise without excluding otherwise eligible participants from a study with a large number of inclusion/exclusion criteria. Previous antibiotic use was not recorded. Importantly, no participants started a course of antibiotics during the study and, as this was a randomised trial, any potential lingering effects of previous antibiotic treatment should be randomly distributed between the Intervention and Control groups. We have added comments to clarify this in Lines 180-184 of the revised manuscript. We should also highlight that three months is a longer exclusion period than some previous studies have used (e.g. PMID: 25021423).

Line 143 - should the limitations of dietary intake measures be reported? Participant characteristics already show an average deficit of 400-500 kcal/day for each group at baseline, which is likely to be due to underestimation of energy intake.

We recognise the limitation of dietary intake measures and highlighted this in the original submission (lines 267-270) with reference to previous research (PMID: 19094249). Due to the inherent limitations of participant-reported dietary intake measures, we based the diet modification on habitual energy expenditure data. This is described in Lines 267-280 of the revised submission.

Line 200 - why was the power calculation driven by changes in fasting plasma, if the aim was to investigate differences in the microbiome? Was this sufficiently powered?

As described in the ClinicalTrials.gov registration (NCT03362554), the primary aim of this trial was to examine the effect of a diet and exercise intervention (that has been previously shown to increase insulin sensitivity) on Rab3 levels and GLUT4 trafficking. Therefore, the power calculation for the trial used fasting insulin as a key measure of metabolic health. This is the key outcome for the present study because we wanted to examine if this change in insulin was accompanied and potentially explained by changes to the gut microbiome and outcomes related to the gut microbiome (e.g., FFAR3 expression). We acknowledge that outcomes related to the microbiome may have inherently different variance and data structures, and thus we have added this as a limitation to the discussion (Revised manuscript Lines 730-736).

Line 356 - were there any dietary controls prior to stool sample collection?

Dietary controls prior to stool sample collection are described (Revised Manuscript Lines 218 and 369). The Intervention group replicated their habitual diet from the preliminary measurement phase, but in reduced quantities. In this way, the diet composition of the Intervention group remained consistent between the baseline and follow-up samples, limiting potential microbiome variation associated with changes in diet composition.

Line 360 - given the manuscript title and aims, why were these participants not excluded from the overall analyses given that this sample is central to the study design?

We feel that it is important to report the effects of the trial on the primary outcome in all participants (fasting insulin) as context for the examined effects on the gut microbiome and measures related to the gut microbiome. As described in the manuscript, faecal samples were not always available and the N for gut microbiome measures was modestly reduced, but the N for other microbiome-related changes was complete (e.g., SCFAs, adipose mRNA).

Line 365 - what was the time until processing and storing at -80?

Samples were processed and stored within 24 h of collection. We have added this to Revised Manuscript Line 379.

Line 488 - Why was the Benjamini-Hochberg method used? While it has been used in studies with larger sample sizes, it has been suggested to be too conservative and likely the reason for studies finding few or no differences between groups (PMID: 30045760)

The ALDEx2 R package (https://rdrr.io/bioc/ALDEx2/f/vignettes/ALDEx2_vignette.Rmd) was used to examine differences in individual taxa between groups and timepoints. This package estimated p values without adjustment and BH adjusted p values. Given the large number of comparisons made, we decided to report the BH adjusted values due to the higher chance of a type 1 error, but in the interest of highlighting potentially important but subtle changes we stated that adjusted p values would be accepted as significant if they were ≤ 0.1 . Balancing type 1 versus type 2 errors is always an important consideration. The raw data from this study is available via repositories for researchers to re-analyse in the event that they wish to use different criteria and/or approaches.

Table 2 - there may be some rounding errors but some of the 'Change' values are different to

the calculated Pre - Post differences reported.

The change values shown in Table 2 are the mean values calculated from the individual change values, and these are not necessarily the same as the simple difference between the baseline and follow-up means. This calculation provides the most accurate reflection of the data, and also permits the calculation of standard deviations of the change values.

Discussion

Line 577 - suggest removing the descriptors 'intensive' and 'vigorous'

We have removed the word 'intensive'. However, we feel that the word 'vigorous' should remain as it is a standard term to define the intensity of the exercise that was employed. Participants exercised at 70% VO_{2Peak} , which meets the ACSM criteria for vigorous intensity exercise (PMID: 21694556).

Line 582 - as above

Please see our previous response.

Line 621 - While Kern et al., did not control for diet, the present study only recorded participant self-reported adherence to their typical diet. The differences in outcome could also be due to differences in the timescale, 16s vs shotgun sequencing, or differences in the statistical approach to the data.

We agree with this point and have added a sentence to highlight other potential differences (Revised manuscript Line 651).

Referee #2:

The study by Thompson et al aims to examine early changes to the microbiome in overweight and obese individuals, using calorie restriction and increased activity as an intervention and compare to a non-intervention group. The authors examine the microbiota via shotgun metagenomics and correlate with tissue and blood markers of energy metabolism as well as physiological indices. Using alpha and beta diversity as the primary measure, they conclude that early weight loss and metabolism is unlikely to be mediated by the gut microbiome. The manuscript is well written and the study is generally well presented. There are some concerns with the approach and design of the study, which may not be sufficient to reach the authors conclusions. I have provided comments below which the authors may wish to consider:

1. The major critique of the study is that the approach to study influence of diet/exercise on the microbiota is rather descriptive and not well defined. For instance, it appears that dietary restriction was approached in a very heterogenous manner, with self-reported and self-managed dietary intervention. This in itself is prone to inaccuracy, but the major issue lies in the lack of data on how the dietary restriction was achieved. One participant may have, for instance, focused on significant reduction of fats, while another focused on carbohydrates while another may have reduced portion sizes. Even then, the habitual diet may vary widely between participants making the impact of these changes difficult to interpret. Without this specific dietary data, it is impossible to uniformly imply that the dietary intervention does or does not impact the microbiota's ecology in a manner that is less obvious than alpha/beta diversity but significant none the less.

We assure the reviewer that the energy restriction was indeed standardised. We did describe the energy restriction intervention in the original submission, but it appears that this was not sufficiently clear and/or detailed. As a result, we have now added a worked example of the calculation. As originally described (Line 282 of the revised submission), Intervention participants were instructed to replicate their 7d weighed habitual food diary for 3 consecutive weeks, but with the recorded weight of each individual food item scaled down to reduce their total energy intake to their individual target value in order to elicit the intended 5,000 kcal/week energy deficit. As a result, participants were not free to change different macronutrients or dietary composition. We regard this as a key strength of our study design as it reduces variance that may result from dietary changes during the intervention. As noted, we have added a new paragraph to the methods to ensure that this is more prominently explained.

2. With this in mind, the authors should provide details on the participants habitual intake and some measurement of participant food intake during the intervention would be advantageous, even a basic breakdown of protein, fats, carbohydrates and fibre intake is likely to be informative, especially when examining changes to the microbiota at a metagenomic level.

Mean (SD) habitual diet data for Control and Intervention groups is reported in Table 1 of the originally submitted manuscript (including total energy intake, carbohydrate, protein, fat, alcohol and fibre).

3. Was hydration/fluid intake over the intervention period measured? Similarly was there any attempt to measure motility/stool frequency? These may impact the study endpoints.

Hydration and stool frequency were not measured during the study period. We have added a note of this in the discussion as a potential limitation (Revised Manuscript Lines 728).

4. A statement on exercise adherence was provided, how did the authors measure diet Adherence?

Participants followed their habitual diet for an extended period (3 weeks) in a free-living setting, and thus it was not feasible to monitor diet adherence over this extended period. As stated in the revised manuscript lines 302, all participants reported following the diet intervention as instructed. It is commonplace for such studies to rely on participant-reported adherence when following an intervention for an extended period in a free-living setting.

5. The second major critique pertains to the study power. The power calculation is based on detecting a reduction in fasting plasma insulin. While this would potentially provide sufficient power for metabolic analyses, it is unclear how this measurement justifies powering the more complex microbiome analyses. Even with that, the study did not analyse data from the required number of participants for the microbiome data (N=23, 9 control, 14 intervention analysed, vs N=24, 8 controls, 16 interventions required). It would be useful if the authors provided an explanation of why the samples for each analysis varies so widely. Figure 1 shows 18 vs 12 for analysis, but this is somewhat misleading for the major endpoints discussed.

As described in the ClinicalTrials.gov registration (NCT03362554), the primary aim of this trial was to examine the effect of a diet and exercise intervention (that has been previously shown to increase insulin sensitivity) on Rab3 levels and GLUT4 trafficking. Therefore, the power calculation for the trial used fasting insulin as a key measure of metabolic health. This is the key outcome for the present study because we wanted to examine if this change in insulin was accompanied and potentially explained by changes to the gut microbiome and outcomes related to the gut microbiome (e.g., SCFAs, FFAR3 expression). We acknowledge that outcomes related to the microbiome may have inherently different variance and data structures, and thus we have added this as a limitation to the discussion (Revised manuscript Lines 730-736).

In terms of the reduced N for faecal samples, as stated in lines 371 of the revised manuscript, some participants were unable or unwilling to provide faecal samples at the required time point.

6. With respect to point 1 and the microbial ecology, have the authors considered analysis of the microbiome at the functional level? The microbial genes related to dietary breakdown and energy harvesting could be assessed and, correcting for the sample drop outs, up/downstream pathways related to the metabolic measurements could be investigated. This would lessen the descriptive nature of the work.

We agree that shotgun metagenomics provides the opportunity to explore further the changes in the metabolic pathways in response to caloric restriction and exercise. Microbiome functional metabolic potential data from the Human Microbiome Project Unified Metabolic Analysis Network 2 (HUMAN2) were analysed. Results from this functional analysis are reported in Figure 3B, and we have also added a new figure (now Figure 6) showing the top 5 changes in individual functional metabolic pathway abundances within and between groups. We observed modest ALDEx2 effect sizes when comparing individual taxa and pathways across all comparisons. The largest effect sizes observed when comparing the intervention group at baseline and follow-up were similar to those observed when comparing the control group at baseline and follow-up. This observation combined with the lack of statistically significant results from our differential abundance analysis (despite accounting for the potential for type 2 errors by increasing the significance threshold to $q < 0.1$) led us to conclude that there were no effects of the intervention on individual taxa or pathways. We could have speculated about potential effects based on those taxa and pathways with the largest observed effect sizes, however, we decided that the data available were not sufficient to support this.

7. The discussion should provide a paragraph on the limitations of the study, particularly pertaining to power.

We have added some additional limitations and considerations to the discussion (Revised manuscript lines 728).

8. It is unclear why the authors did not include individual intervention groups (diet or exercise) in the experimental design, as the current design would make it impossible to ascertain whether one intervention dominates or is even required.

We agree that additional intervention arms could have been interesting, but the purpose of the current study was not to examine the effects of exercise and caloric restriction in isolation. We used a combined intervention that we previously showed led to metabolic changes of the type required to examine microbiome-related outcomes. It would have doubled the size of the study to include additional arms, but it is not clear how this would have helped us to address our primary question. With the current design, if we had seen interesting temporal relationships, then future studies could have sought to disentangle the nature of these relationships and effects.

Referee #3:

Summary:

This study investigated whether a 3-week combined exercise and dietary energy restriction intervention in sedentary adults with overweight/obesity leads to changes in the gut microbiome, potentially mediating improvements in metabolic health. Despite significant reductions in body mass, fat mass, and improvements in insulin sensitivity, no significant

changes in gut microbiome diversity or composition were observed. The study is methodologically sound and addresses a relevant question, though the interpretation of results and presentation of microbiome data would benefit from some revisions to strengthen the manuscript's impact and clarity.

Strengths:

1. **Robust Study Design:** The randomized controlled trial (RCT) design strengthens causal inference, addressing a critical gap in existing literature that mostly comprises observational Studies.
2. **Comprehensive Methodology:** Use of shotgun metagenomics (Kraken2/Bracken and HUMAnN2) provided in-depth analysis of gut microbiome changes.
3. **Multi-Tissue Analysis:** The study uniquely measured SCFA concentrations and gene expression related to host-microbiome interactions in both skeletal muscle and adipose tissue.
4. **Clear Presentation of Results:** The manuscript presents results concisely with appropriate statistical analyses, and figures/tables are well-structured.

Thank you for your positive comments.

Weaknesses and Areas for Improvement:

1. Short Intervention Duration:

- o A 3-week period may be insufficient to observe gut microbiome changes. Longer interventions could reveal delayed microbial adaptations.
- o Recommendation: Discuss limitations of the short duration and suggest future studies with extended timelines. Include the acute nature of the intervention in statements such as "...the human gut microbiome remains stable in the face of an intensive energy restriction and vigorous exercise intervention".

We agree that we can only comment on short-term changes and have now added "short-term" in several places where we describe the intervention (for example, lines 595, 600, 666, 738). We have also added a comment to the discussion regarding the implications for shorter and longer-term effects (Line 662 of the revised manuscript). We do not see the short-term nature of the intervention as a limitation for interpretation of the present findings – this design and time scale allows us to test a specific question.

2. Sample Size and Power Calculation:

- o Although justified, the modest sample size (n=30) may limit detection of subtle microbiome shifts.
- o Recommendation: Include a discussion on whether the study was sufficiently powered to detect microbiome changes.

We have made some additions to the discussion to highlight this as a potential limitation (Revised Manuscript Line 730).

3. Dietary Composition Control:

- o The study maintained participants' habitual diets but only reduced energy intake. Specific dietary components influencing the microbiome (e.g., fiber, prebiotics) were not controlled.
- o Recommendation: Discuss how unaltered diet composition might limit microbiome Responsiveness.

We view the maintenance of habitual dietary composition throughout the intervention as a strength of the study design. It reduces potential variance in microbiome composition that

may be attributed to dietary composition changes and improves our ability to isolate the effect of combined exercise and energy restriction. It is possible that if we had altered dietary composition then this could have produced changes in microbiome-related measurements, but this would have addressed a different question.

4. Microbiome Data Presentation:

o Alpha and beta diversity metrics are reported, but taxonomic and functional pathway data are underexplored.

o Recommendation: Consider providing more detailed analyses of specific taxa or pathways potentially affected by the intervention.

We have added a new figure (now Figure 6) showing the top 5 changes in individual functional metabolic pathway abundances within and between groups. We observed modest ALDEx2 effect sizes when comparing individual taxa and pathways across all comparisons. The largest effect sizes observed when comparing the intervention group at baseline and follow-up were similar to those observed when comparing the control group at baseline and follow-up. This observation combined with the lack of statistically significant results from our differential abundance analysis (despite accounting for the potential for type 2 errors by increasing the significance threshold to $q < 0.1$) led us to conclude that there were no effects of the intervention on individual taxa or pathways. We could have speculated about potential effects based on those taxa and pathways with the largest observed effect sizes, however we decided that the data available were not sufficient to support this.

5. Interpretation of Null Findings:

o The conclusion that early metabolic improvements are independent of microbiome changes is strong but may overlook other microbial functions not captured by the methods used.

o Recommendation: Use hedging language (e.g., "may not be mediated") and discuss alternative mechanisms. Furthermore, include the acute nature of the study in the language that "metabolic improvements are independent of microbiome changes".

We have softened statements in the discussion to make them less definitive.

Minor Issues:

- Grammar and Clarity: Minor grammatical improvements could enhance clarity, particularly in the Discussion section.

- Figure Legends: Ensure all figures and tables are fully self-explanatory with clear legends.

We have checked the discussion and figure legends for clarity and accuracy.

END OF COMMENTS

Dear Professor Thompson,

Re: JP-RP-2025-287424R1 "Effects of a combined energy restriction and vigorous intensity exercise intervention on the human gut microbiome - a randomised controlled trial" by Dylan Thompson, Russell Graham Davies, Laura A Wood, Ciara O'Donovan, Wiley Barton, Fiona Crispie, Jean-Philippe Walhin, Maria A Valdivia-Garcia, Gary Frost, Orla O'Sullivan, Paul D Cotter, Javier T Gonzalez, James A Betts, Françoise Koumanov, and Aaron Hengist

We are pleased to tell you that your paper has been accepted for publication in The Journal of Physiology.

Yours sincerely,

Kim Barrett
Senior Editor
The Journal of Physiology

If you would like to receive our 'Research Roundup', a monthly newsletter highlighting the cutting-edge research published in The Physiological Society's family of journals (The Journal of Physiology, Experimental Physiology, Physiological Reports, The Journal of Nutritional Physiology and The Journal of Precision Medicine: Health and Disease), please click this link, fill in your name and email address and select 'Research Roundup':
<https://www.physoc.org/journals-and-media/membernews>

- You can help your research get the attention it deserves! Check out Wiley's free Promotion Guide for best-practice recommendations for promoting your work at: www.wileyauthors.com/eeo/guide. You can learn more about Wiley Editing Services which offers professional video, design, and writing services to create shareable video abstracts, infographics, conference posters, lay summaries, and research news stories for your research at: www.wileyauthors.com/eeo/promotion.

EDITOR COMMENTS

Reviewing Editor:

The authors have responded appropriately to each of the Reviewer's concerns with the result that this interesting and well-conducted study is improved over the original submission.

Senior Editor:

Please accept our apologies for the unfortunate delay that occurred in rendering a final editorial decision on the very nice revision of your manuscript.

REFEREE COMMENTS

Referee #2:

I thank the authors for their considered revisions. The manuscript is far clearer and I have no further comments.

Referee #3:

Thank you very much for this responsive resubmission. I have no further comments.